# Laboratory and Field Test and Distinct Element Analysis of Dry Granular flows and segregation process

Cheng Yung Ming[1,3], Fung Wing Hong Ivan[2], Li Liang[3] and Li Na[1]

Department of Civil and Environmental Engineering, The Hong Kong Polytechnic University, Hong Kong[1]

Department of Architecture and Civil Engineering, City University of Hong Kong[2]

School of Civil Engineering, Qingdao University of Technology[3]

## Abstract

Natural as well as fill slopes are commonly found in Hong Kong, China and many other countries, and slope failures with the subsequent debris flows have caused serious loss of lives and properties in the past till now. There are various processes and features associated with debris flow for which the engineers need to know so as to design for the precautionary measures. In this study, experiments on flume tests, friction tests, deposition tests, rebound tests have been carried out for different sizes of balls to determine the parameters required for the modelling of dry granular flow. Different materials and sizes of balls are used in the flume tests, and various flow pattern and segregation phenomenon are noticed in the tests. Distinct element (DEM) dry granular flow modeling are also carried out for the flow process. It is found that for simple cases, the flow process can be modelled reasonably well by DEM which is crucial for engineers to determine the pattern and impact of granular flow which will leads to further study in more complicated debris flow. From the laboratory tests, large scale field tests and numerical simulations of the single and multiple material tests, it is also found that the particle size will be the most critical factor in the segregation process during granular flow. It is also found from the laboratory tests and numerical simulations that a jump in the flume can help to reduce the final velocity of the granular flow which is useful for practical purposes.

Keywords: flume test, field test, balls, granular flow, distinct element, flow process

## 1- Introduction

The terrain of Hong Kong is hilly. Many slopes (fill, cut and natural slopes) and roads are formed to cope with the rapid development of Hong Kong, China and many other developed cities. Hong Kong has a high rainfall, with an annual average of 2300mm which falls mostly in summer between May and September. The stability of man-made and natural slopes is of major concern to the Government and the public. Landslides and the subsequent debris flows have caused loss of life and significant amount of property damage in the past. In Hong Kong, for the 50 years after

1947, and more than 470 people died due to slope failures and debris flow associated with man-
made cut slopes, fill slopes and retaining walls.
There are many reported serious slope failures and debris flow problems in China in the recent ten
years, due to the significant amount of constructions and inadequate stabilization to many
temporary or permanent fill or natural slopes. The destructive power of large scale debris flow is
well known, and the prevention of slope instability, reduction of debris flow destructive power by
the use of rigid, flexible barrier or other means are well practiced in many countries. There are
many cases where the slopes fail with subsequent debris flows in Hong Kong and China (Scott
and Wang 1997), which have created various serious problems. Based on a conservative estimate,
over 60 countries in the world have faced the problems of debris flow over the years. With
reference to Fig.1, the debris flows in Hong Kong and China have created traffic problems, serious
loss of lives and properties, and currently there are many active research works in the area of debris
flow in Hong Kong and China. The research works include three-dimensional slope stability
analysis, debris flow process, impact loads on flexible and rigid barriers and others. An example
on three-dimensional slope stability analysis using 16000 columns has been carried out by Cheng
in 2016/2017 which is shown in Fig.2a (Lo et. al. 2018). The analysis of the non-spherical surface
is achieved by the use of Nurbs function as discussed by Cheng et al. (2005). Upon the
determination of the critical failure mass, ~~and~~ the flow path of the soil can be estimated from a
distinct element analysis using the method as discussed by Cheng et al. (2015). The slope failure
and the subsequent debris flow ($2100m^3$ of debris) as shown in Fig.2b is finally protected by the
use of three levels of flexible barrier against the future potential debris flow. The authors are also
considering the use of meshless method in the assessment of debris flow, which will be the next
stage of the present work (Wong 2018).

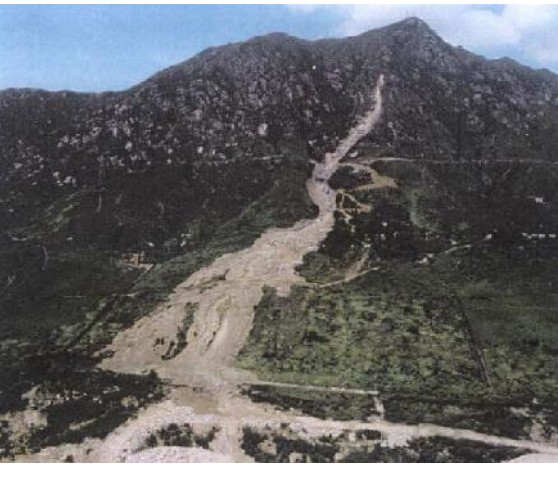   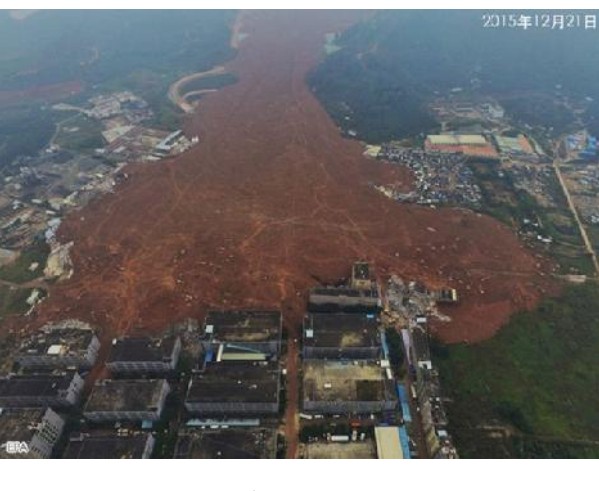

(a)                                              (b)

Fig.1 Representative debris flow in Hong Kong and Shenzhen, China (a) Tsing Shan debris flow
in 1990 (King 2013); (b) debris flow in Shenzhen 2015 (see Wikipedia).
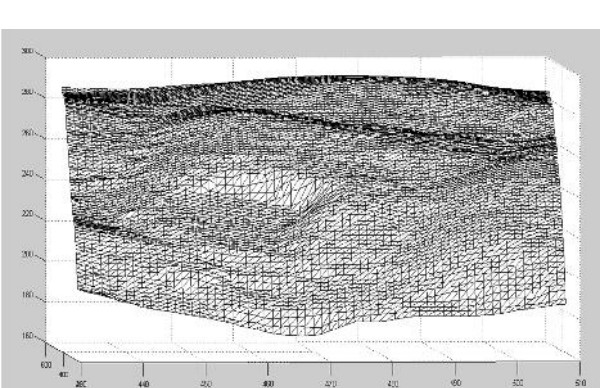 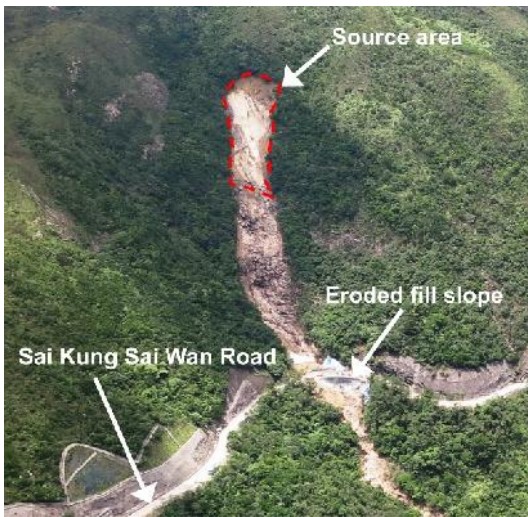

(a) 3D slope stability analysis        (b) Debris flow after slope failure

Fig.2 Three-dimensional slope stability analysis by Cheng (the triangulation represent the
geometry as defined by the GIS information) and the subsequent debris flow for a slope in Hong
Kong has blocked the Sai Wan Road traffic

Granular flow as a pilot study of debris flow has some fundamental difficulties in the physical tests
as well as numerical analysis. In general, various particles sizes will be present in a flow, and the
debris mix is usually far from uniform in composition. For physical tests, it is difficulty to apply a
representative debris flow mix, and the flow process is further complicated by the presence of
water. For numerical simulation, it is virtually impossible to accommodate too much particles in a
model, ranging from a very small particle size to cobbles or even boulder in the extreme range.
Even if such a numerical model can be established, there will be serious numerical problems if the
particles sizes differ too much in the system. Granular flow can be induced from gravity, driven
by fluid dynamic or from both factors. The classification of debris has been given by Varnes (1978),
and later modified by Furuya (1980), Ohyagi (1985), Pierson and Costa (1987), Coussot and
Meunia (1996), Cruden and Varnes (1996), Hungr et al. (2001), Takahashi (2001, 2006) and others.
A detailed theoretical treatment of dry granular flow similar to some of the single material tests in
the present study has been given by Takahashi (2014) and will not be repeated here. In this study,
we will concentrate mainly on the action of gravity, while the effects of water is under study by
the authors as the next stage of research work.
Many scientists have carried out granular flow analysis. Lo (2004) ~~has~~ compared ~~the~~ different
composition of granular flow in landsides in Hong Kong and examined the coarse and fine particle
concentration. Hutter et al. (2005) ~~has~~ considered the flow envelops and the deposition of the flow.
In year 1991, the U.S. Geological Survey has made a large scale flume for detailed experimental
tests on debris flows. Mizuyama and Uehara (1983) ~~have~~ made a flume which is 20 cm wide and
25m long, and the slope angle ranged from 5 degree to 25 degree. Liu (1996) ~~has~~ made a 18 cm
depth, 16 cm width and 150 cm length flume in Yunnan, China, and the flume inclination can be
adjusted from 10 to 34 degrees. Lin (2009) ~~has~~ made a 20 cm width 8m length flume with a 2.2 m
width 3 m length catchment. There are also various flume tests that have been carried out by
various researchers in Hong Kong and many other countries.
During the transportation period, segregation occurs when debris starts to flow. Iverson (1997)
studied the factors that influence the segregation process. He found that particle size has a great
effect on the segregation process, and debris with larger particle size move upward while fine
particles go downwards. This phenomenon is the opposite of "normal grading" in which the finer
particles are found at the upper layers in the lake or river and large particles rest at the bottom. The
main reason for the segregation is kinetic sieving, and finer particle can go through the gaps
between particles more easily than the larger particle. Large particles can also be found at the front
of the flow because of the relatively high velocity of the larger particles at the upper layer,
compared with the finer particles with lower velocity at the lower layer. When a stable contact
network for large particle is formed at the free surface, the segregation cease to occur and the balls
finally deposit at the catchment area.
For distinct element modeling (DEM) of granular flow, Jiang et al. (2003) ~~has~~ studied the methods
of generations of ball in PFC2D (Cundall 1971, 1988, Cundall and Hart 1992, Cundall and Strack
1979), namely the expansion method and isotropic compression method. Zohdi (2007), Halsey and
Mahta (2002) discussed about the physics of granular flow; the contact model and the limit of the
friction coefficient. Sullivan (2011) also compared the theory and computation in distinct element
analysis. It is well known that the use of DEM can only provide qualitatively instead of quantitative
study up to the present (see also the discussion part), and most researchers adopt DEM for
qualitative analysis only.
In the present study, dry granular flow experiments will be conducted under different conditions
using glass and rubber balls for a basic study on the flow process and segregation. Both glass and
rubber balls of different diameters have been used in the tests, and combination of different size
and materials have also been tried in the tests for the illustration of the segregation problem. The
experimental results are also analyzed by distinct element analysis using program PFC2D. It is
true that three-dimensional distinct element modelling can be a better tool for the present problems,
but the previous experience in three-dimensional distinct element modelling by the authors suggest
that the amount of computer time can be significant. For the present study, the flume in both the
laboratory and field tests are relatively narrow, and off-track movement of the balls/grains are not
major. In view of that, two-dimensional modelling has been adopted in the present study, and good
results are actually obtained. The tests are performed at relatively simple condition so that the basic
problem of flow and segregation can be studied easily. It should also be mentioned that more than
10 ten thousands photos are taken from the laboratory and field tests, and such amount of
information cannot be fed into a paper. In views of that, only representative intermediate photos

128 which are used for illustration are given in the present paper, while some of the observed
129 phenomena are simply description without the support of the photos.


131 **2. Physical flume modeling of granular flow**

132 **2.1 Instrumentation and Test Material**

133 To enhance the knowledge on the granular flow mechanism, many laboratory and large scale field
134 tests have been carried out by the authors. The laboratory model is about 1.5m long and 1.3m high
135 (adjustable). The flume in the laboratory is made of polystyrene and is designed to be flexible, and
136 the angle of inclination can be adjusted if necessary. The flume model is 40cm depth, 40 cm width,
137 140 cm length of upper flume and 100 cm for the lower flume with a 60 x 60 catchment area at
138 the bottom. Fig. 3 and Fig 4 show the schematic design of flume and flume model in the laboratory
139 tests. In order to record the motion of the particles, two high speed cameras are adopted. The first
140 one is mounted on the upper flume while the second one is fixed to the bottom flume. In the
141 laboratory tests, different sizes of glass beads and rubber beads are used to replace the use of sand,
142 and this simplification can help to assess the effects of shape and material on the segregation
143 process. In the large scale field test, real sand is used. For the material parameters, the dynamic
144 friction angle is measured by using tilting test (Pudasaini & Hutter (2007), Mancarella & Hungr
145 (2010)). The property of the glass and rubber beads are determined experimentally, and the details
146 are given in Table 1.

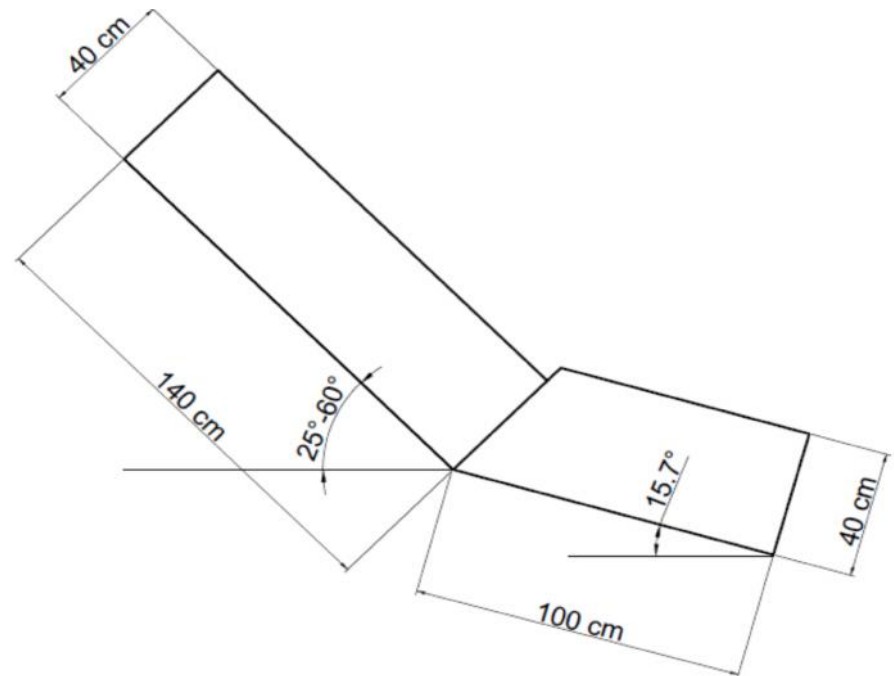


Fig.3 Schematic Design of Flume


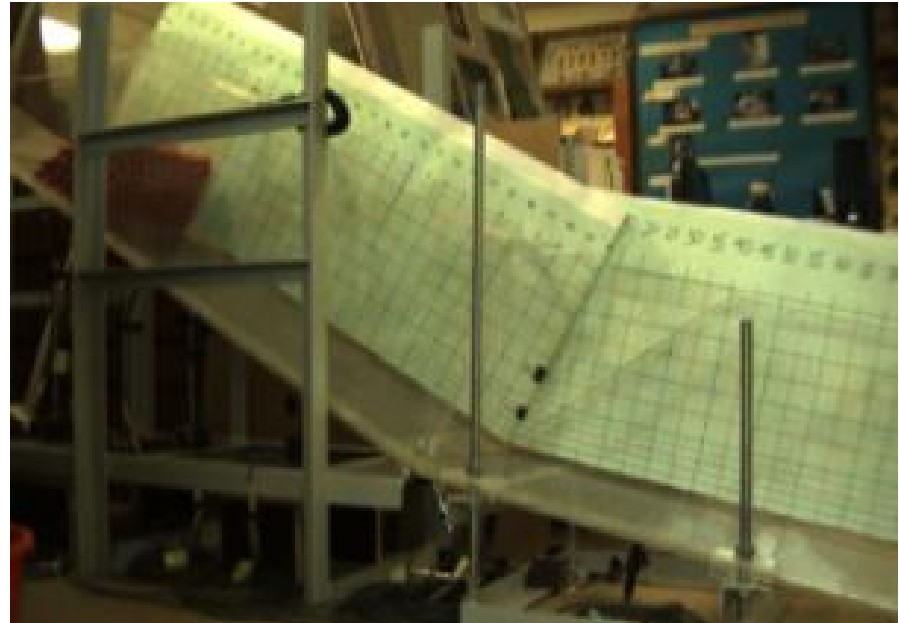


Fig.4 Flume model in laboratory

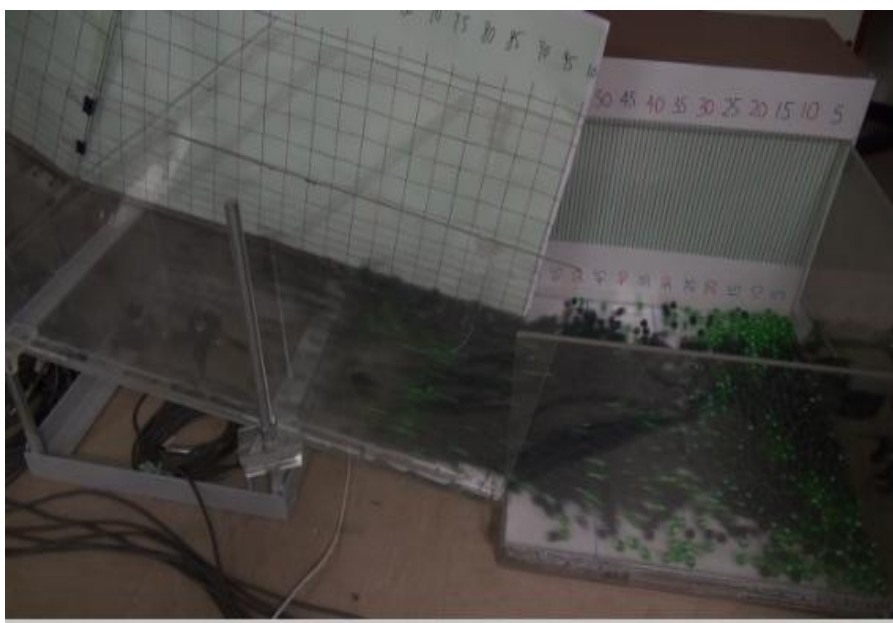


Fig 5.  Flume model with a small jump in laboratory


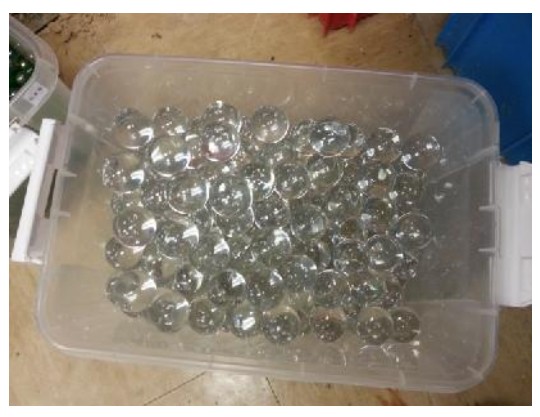 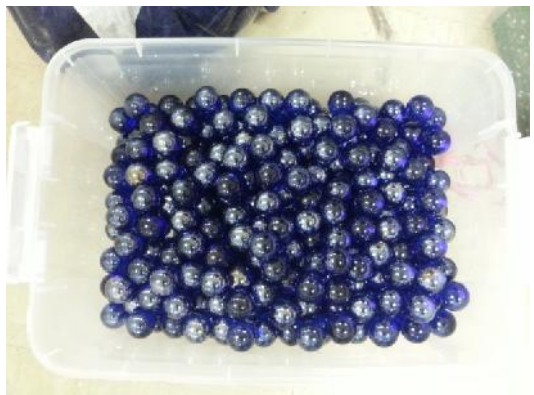

Fig.6a Transparent glass                    Fig.6b Blue glass ball

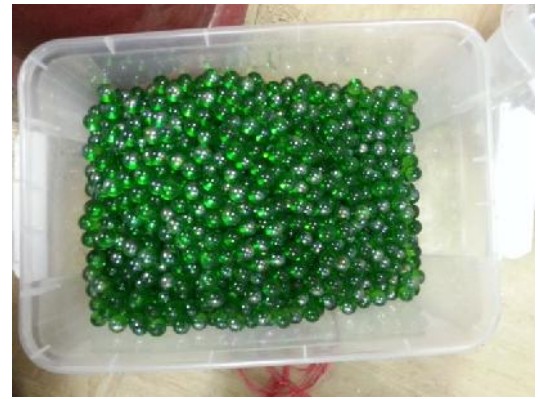 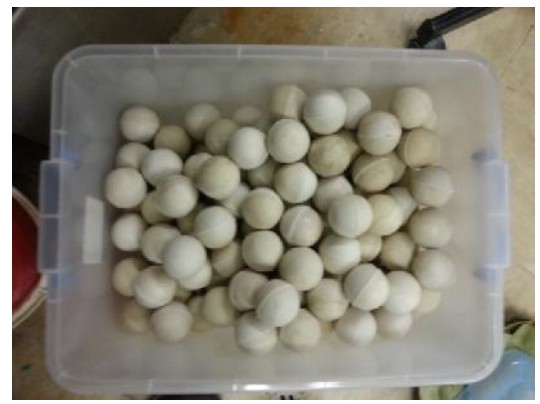


Fig.6c Green glass ball                      Fig.6d White plastic ball

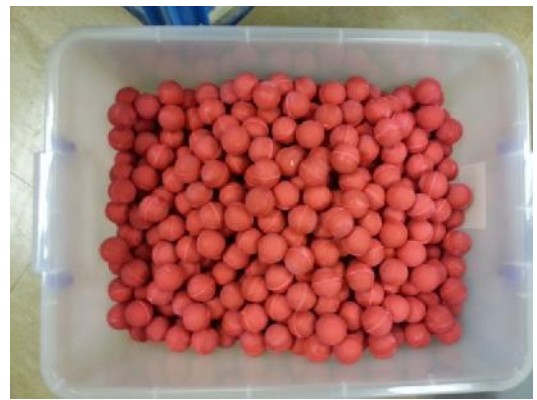 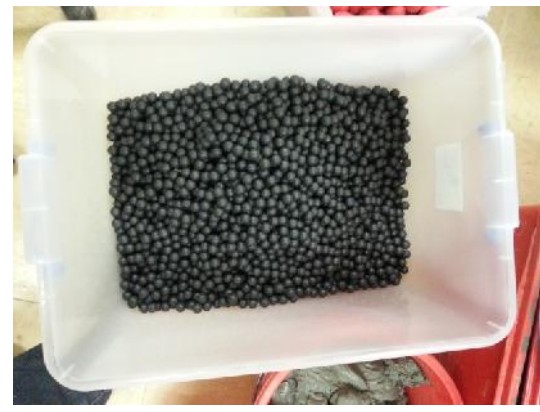


Fig.6e Red plastic ball                      Fig.6f Black plastic ball

Table 1.  The properties for the glass balls and plastic balls in laboratory granular flow test

| Plastic | D(mm) | Average Weight | Density (kg/ m$^3$) | External Friction Coefficient | Internal Friction Coefficient |
|---|---|---|---|---|---|
| White | 50 | 105.35 | 1609.64 | 0.781 | 0.547 |
| Red | 30 | 23.382 | 1653.97 | 0.630 | 0.429 |
| Black | 15 | 2.862 | 1619.56 | 0.222 | 0.365 |
| **Glass** | D(mm) | Average Weight | Density (kg/ m$^3$) | External Friction Coefficient | Internal Friction Coefficient |
| Transparent | 40 | 78.686 | 2348.11 | 0.102 | / |
| Blue | 25 | 21.121 | 2581.64 | 0.053 | / |
| Green | 16 | 5.744 | 2678.28 | 0.104 | / |


## 2.2 Test Programme


In the present study, the angle of the flume in laboratory is kept to be 45 degree. The effect of the
slope inclination will not be discussed in this paper, but the test results by the authors show that
the segregation process will basically remain unchanged with different flume inclination. The
effect of flume inclination can affect the degree of segregation as well as impact forces which will
be covered by a separate paper later. Totally 68 laboratory tests have been carried out. The 68 tests
are divided into two groups: the first group of tests were conducted on the flume with a small jump,
and the other group of tests were carried out on the flume without a jump. Such a jump is also
commonly adopted in Hong Kong, and this helps to lower the velocity of the granular flow (for
small scale flow). Fig 5 shows the flume in laboratory with a small jump. The effects of the particle
size and the flowing mass are also studied through the use of balls with different diameter, mass
and combination of different balls. Table 2 shows only some of the test programme. Test 1 to test
48 belong to the first tests group with a small flume jump. Test 1 to test 6 were carried out by using
six different kinds of balls separately with the same mass of 10 kg. The mass of the balls is then
changed to 13.55kg and the above tests are repeated again (for test 7 to 10). In order to study the
segregation process for test 11 to 40, two kinds of balls with different diameters were combined
together, and for the same purpose in test 40 to test 48, three kinds of balls were combined together.
Test 49 to test 68 belong to the group without a small flume jump. Same as the first group of tests

with a small flume jump, test 49 to test 55 were carried out for same material but different sizes of balls. In test 56 to test 63, combinations of two kinds of balls were tried. The last five tests were the combination of three kinds of balls.

Table 2.  Test Programme

| Flume with a small jump | | | | |
|---|---|---|---|---|
| One kind of balls | Test Number | Flow Mass | | Balls |
| | 1 | 10 Kg | | G(Transparent) |
| | 2 | 10 Kg | | P(White) |
| | 7 | 13.55Kg | | G(Green) |
| | 8 | 13.55Kg | | P(Red) |
| Two kinds of balls | Test Number | Top Layer | | Bottom Layer |
| | 11 | P(White) | | P(Red) |
| | 26 | G(Trans) | | P(White) |
| Three kinds of balls | Test Number | Top Layer | Middle Layer | Bottom Layer |
| | 41 | P(White) | P(Red) | P(Black) |
| | 45 | G(Trans) | P(Red) | P(Black) |

| Flume without a small jump | | | | |
|---|---|---|---|---|
| One kind of balls | Test Number | Flow Mass | | Balls |
| | 49 | 10 Kg | | G(Transparent) |
| | 50 | 10 Kg | | G(Blue) |
| Two kinds of balls | Test Number | Top Layer | | Bottom Layer |
| | 55 | P(White) | | P(Black) |
| | 56 | G(Trans) | | P(Black) |
| Three kinds of balls | Test Number | Top Layer | Middle Layer | Bottom Layer |
| | 67 | G(Trans) | P(Red) | P(Black) |
| | 68 | G(Trans) | P(Red) | G(Green) |

P: P refers to plastic balls, G: G refers to glass beads

## 2.3 Test procedure and test results

Test materials with different particle size combinations (single type of balls to multiple types of balls) were put into the container which is on the top of the flume. Figure 7 shows the flow pattern of single type dry granular material flowing along the flume. The video captured by high speed camera can show this process clearly. When the gate of the container was pulled up, the front part of flow mass become loose and start to flow along the upper flume under the action of gravity, while the latter part of flow mass followed behind. Flow mass elongated when it moved forward,

and the shape of flow front is wedge-like type. At the moment when the particles reached the
bottom of the flume, the velocity direction of the balls changed because of the angle difference
between the upper flume and the lower flume. During the transportation period, a large amount of
potential energy of the initial flow mass was transferred to momentum energy accompanying by
energy dissipation through the grain collision and friction. Particles at the front of the flow
reflected back when they impacted on the wall of deposition zone and collided with the subsequent
particles immediately, which consumed the residual momentum energy of flow particles. Finally
all the particles rested in the deposition zone.
In reality, there are sediments and water in a debris flow. The effect of water is complicated and
will not be studied in the present work. The grain size distribution is usually not uniform as in the
present laboratory tests. Consequently, a good understanding of the particle flow under a mixture
of ball sizes is important. Particle size is a vital parameter for the good understanding of multi-size
particle flow because it not only has an effect on the flow dynamic, but also influence the energy
attenuation during the whole flow process. Furthermore, the tilting test that is mentioned above
demonstrates that the dynamic friction angle depends on the particle size, specifically, larger
particle size will has smaller dynamic friction angle while smaller particle size will has larger
dynamic friction angle. The flow pattern of multi-size particle flow is more complicated when
compared with the single size particle flow.
Figure 8 shows the flow pattern of multi-size particle flow. Segregation occurred when the
combined particles started flowing along the flume. Figure 8a demonstrates the flow pattern of
multi-size particle flow composing of white and black plastic balls. The diameter of the white
plastic ball is much larger than the black plastic ball as shown in Table 1. From the video captured
by the high speed camera, it is easy to observe that during the transportation period, white plastic
balls flowed on the upper layer while black plastic balls stayed at the bottom layer. This
phenomenon is consistent with the segregation theory of Savage et al. (1988). Besides, it is not
difficult to find that white plastic ball always stayed at the front of the flow where the velocity was
the highest, in other word, the velocities of the white plastic balls with relative larger diameters
are higher than the black plastic balls. Besides, at the upper layer where larger white plastic balls
are located, the inertial force dominated the flow dynamic and the energy dissipation was less than
that of the lower layer where the flow motion is mainly controlled by the contact forces. For the
forgoing reasons, it can be seen that large particle size leads to higher velocity during the flow.
Figure 8b shows the flow pattern of multi-size material composing of green glass balls and black
plastic balls. The diameter of green glass ball is similar to the diameter of black plastic ball, while
the density of green glass ball is almost two times larger than black plastic ball. In the upper
container, green glass balls were put statically at the top of the black plastic balls. After pulling up
the door, the black plastic balls flowed out firstly at the beginning and stayed at the bottom layer
due to the arrangement of the initial position of balls in the container, green glass balls quickly
moved downwards under the action of gravity, which leads to the green glass balls at the upper
layer replaced by black plastic balls subsequently. When the black plastic balls form a stable
contact network at the upper layer of the flow, the position transition or segregation process
stopped. In this case, the difference of particle sizes between two kinds of balls is not obvious, and
segregation was initiated due to the density difference only. During the segregation process in
which green glass balls moved downwards and black plastic balls migrated upwards, the
momentums of these two kinds of balls were transferred to each other at neighbor location,
therefore green glass balls and black plastic balls arrived at the catchment area almost at the same
time, while for the test in which balls were arranged in an opposite order (black plastic balls at top
and green glass balls at bottom), the green glass balls move faster and deposit earlier at catchment
area compared with the black plastic balls due to the smaller dynamic friction angle as well as the
larger kinetic energy of the green glass balls.
Similar to the above two figures, Figure 7c shows the flow pattern of transparent glass balls and
black plastic balls. In this case, both the density and particle size of the transparent glass balls are
larger than that of the black plastic balls. As shown in high speed camera video, during the flow
process, the transparent glass balls flow upwards and move faster in comparison with the black
plastic balls. Hence, although the density of the transparent glass balls is larger than the black
plastic balls, the transparent glass balls still stay at the upper layer of the granular flow due to their
relatively large particle sizes, which means that particle size has greater contribution for the
segregation process than density in the analysis of granular flow.

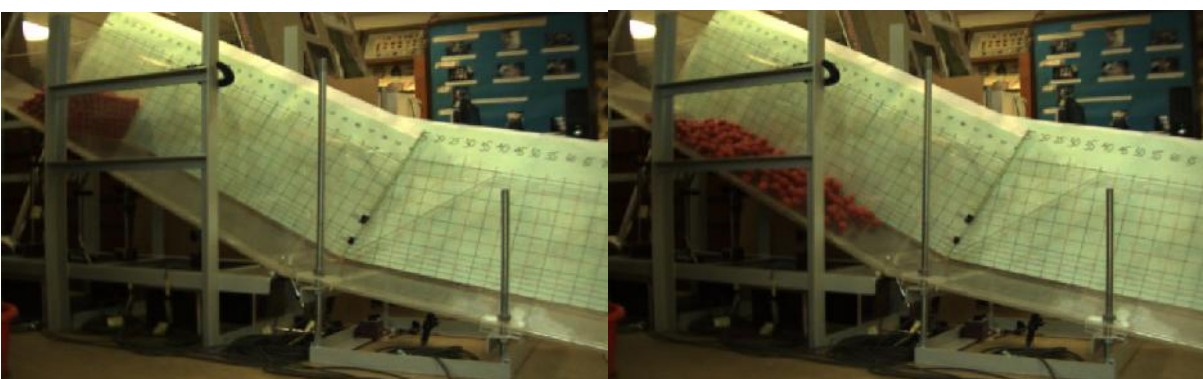


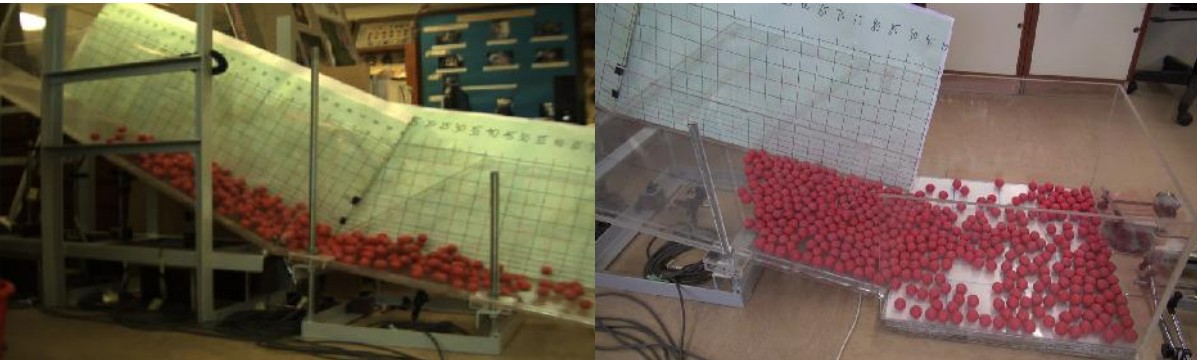


Fig. 7. Flow pattern of mono-size particle flow in physical model

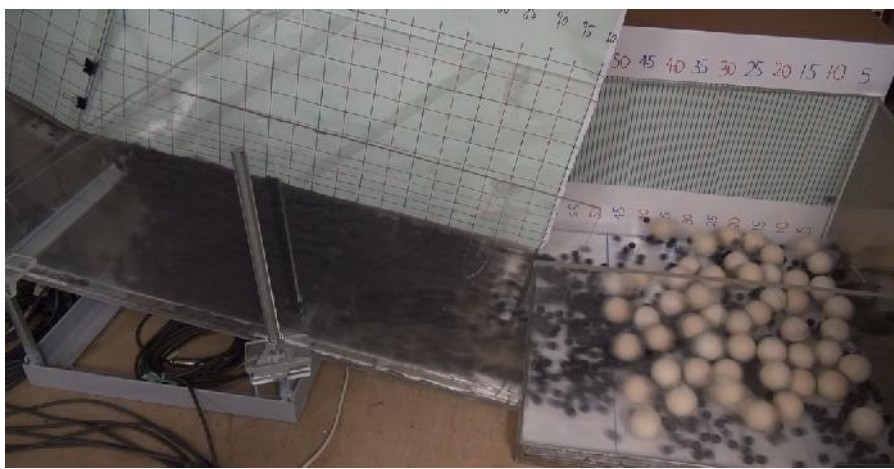


a)  The influence of particle size on segregation process

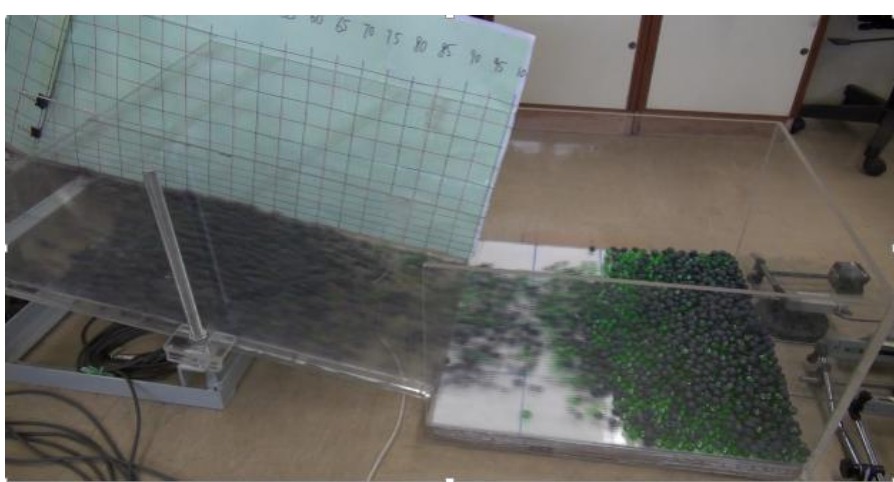


b)  The influence of density on segregation process

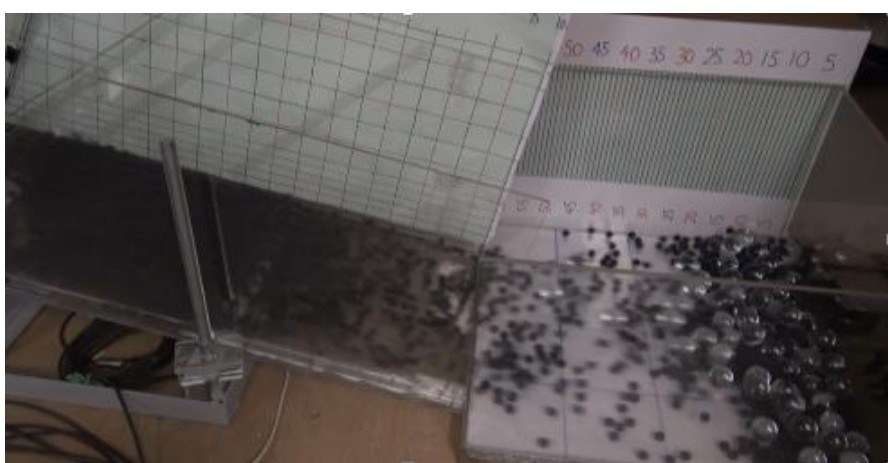


c)  The influence of particle size and density on the segregation process
Fig. 8. Flow pattern of multi-size particle flow

## 3. Numerical Modeling of granular flow

### 3.1 Model generation

Previous model tests by Chan (2001) for the runout were calibrated by the Dan model, where the
problem of segregation and flume jump were not considered. In general, the results are in
agreement with those from Rickenmann (in Jackobs and Hungr 2005). For the present studies
where multi-size particles are considered, the use of the simple Dan model is insufficient. The use
of meshless method to model debris flow has recently been considered by the authors (Wong 2018).
While the meshless method can give a prediction of the debris flow process, the segregation
phenomenon is totally neglected in the analysis, but such phenomenon is found to be critical for
many cases in Hong Kong. In views of the limitations of these numerical methods, the laboratory
tests in the present study are modelled using the distinct element method, which is more
appropriate for the large deformation, segregation and separation phenomenon during the
transportation process. Once the appropriate numerical model is established, the numerical
technique will be extended to the field tests for which natural sand is adopted. In this paper
commercial program PFC2D using DEM has been adopted to implement the numerical simulation
of dry granular flow. Totally, there are five different methods of model generation in PFC2D
program, and based on the consideration of time requirement, the rain method was adopted finally.
The parameters used in the numerical simulation are the micro-properties which are difficult to be
determined. Benchmark tests have been carried out in order to calibrate the micro-mechanical
properties of the dry granular material. Some of the micro-parameters of the balls are determined
through changing their values so that the macroscopic behaviors in numerical simulation are
consistent with that in physical test. The detailed micro-properties of the balls are shown in Table
3. Except for the wall friction (should be small as the walls are relatively smooth) and wall stiffness,
all the other parameters in Table 3 are determined by laboratory tests. In order to get different
frictional coefficients among the balls, two piece of wood which have plastic balls stick on it
regularly and shear force is applied. Furthermore, depositional tests, rebound tests are carried out
to measure the frictional angle and rebound coefficients of the balls. For each parameter, five
laboratory tests have been carried out, and the mean values are presented in Table 3. It should be
noted that there is not a wide distribution in the laboratory determined parameters, hence the range
of these parameters are not shown for clarity. The diameters of the particles in the numerical
analysis are the same as that used in the physical tests.

Table 3. Microscopic parameter of the balls for granular flow analysis

| Balls | Ball stiffness $(N/m^2)$ | Ball damp | Ball density $(kg/m^3)$ | Ball friction | Wall friction | Wall stiffness $(N/m^2)$ |
|---|---|---|---|---|---|---|
| Red plastic ball | 2.36e9 | 0.4 | 1250 | 0.462 | 0.1 | 1.11e11 |
| Black plastic ball | 7e8 | 0.2 | 1250 | 0.1 | 0.1 | 1.11e11 |
| Blue glass ball | 7e10 | 0.3 | 2500 | 0.1 | 0.1 | 1.11e11 |
| Green glass ball | 7e10 | 0.2 | 2500 | 0.1 | 0.1 | 1.11e11 |

## 3.2 Numerical test results

A detailed comparison of the granular flow pattern modeled by the physical tests and discrete element analysis is shown in Figure 9. Figure 9a shows the physical test in which both the red plastic balls and green glass balls were used (too many test results are available, and only selected results are used for illustration in this paper). Large blue balls and small red balls in the numerical model represent the actual red plastic balls and green glass balls in the physical model tests respectively. A full-scale numerical simulation is rare to be conducted for discrete element analysis due to the limitation of the computer resource, but this is considered to be necessary and acceptable for the present study. Figure 9b shows the numerical results of the flow pattern of the multi-size particles. Particles start to flow along the flume after the initiation of the flow. During the flow process, the flow mass became longer under the action of shear force. Particles moved apart from each other and pushed other particles forwards. During this process, the momentums of the balls were exchanged and transferred to other balls at the neighbor locations. The flow velocity keep increasing until the front of the flow hit on the wall of the deposition zone. When the kinetic energy of the balls was exhausted, the balls eventually ceased to move at the catchment area. Figure 10 shows the flow pattern of multi-size balls flows composing of black plastic balls and green glass balls of which the diameter are relative smaller than the other balls as considered in the present paper. A pronounced Saltation was observed as balls flowed, implying that the collisional character of the flow mass where the savage number is larger than 0.1 (if the savage number is smaller than 0.1, the flow belongs to frictional flow, Iverson 1997). Savage number is the ratio between inertial force and frictional force. The comparison between Figure 10 and Figure 9b indicates that the larger the ball size, the more collisional the flow mechanism would be. As a result, the inertial forces dominate the flow dynamic compared with the frictional forces in the present tests. Furthermore, the balls at the upper region of the flow associated with higher velocity had more

collisions and moved freely compared with that at the bottom region. The balls at the lower region
were compacted with lower flow velocities. By comparison, the numerical simulation results of
the flow pattern have a very good agreement with the physical test results when the micro-
parameters were selected suitably.
As shown in Figure 9b and Figure 10, segregation was also observed in the numerical model after
the dry granular balls started to move. In Figure 9b, it was evident that the blue balls with larger
ball size moved upwards and forwards, while the red balls with smaller ball size went to the lower
layer and stayed at the rear of the flow, which was consistent with the results in the physical model
tests. Smaller particles are more likely to move through the void between the larger particles, and
this will in turn squeeze the large particles to the upper layer of the flow. Because of the momentum
exchange between the balls and the flow mass dilation resulting from the shear deformation, a
dispersive pressure was caused which result in larger dry granular balls moved faster than the finer
particles and went upwards, and lead to the results that larger balls flowed to the upper layers
where the shear strain is low and accumulated at the front of the flow, while the finer balls tend to
moved downwards and accumulated at the bottom of the flow (Takahashi (1981)). Besides, the
difference of the ball size induce an unbalance forces on the balls which restrict the vertical
movement of the balls, this will also affects the flow segregation in the vertical direction.
Furthermore, the density difference between the balls the in numerical model is another factor that
influence the segregation process. Particles with lower density are more likely to rise to the free
surface while particles with higher density are more likely to segregate to the bottom of the flow.
From Figure 5b, it can be noticed that it is easily for the red balls with larger density traveled
through the gap generated by the shear deformation and squeezed the particle with smaller density
up to the upper flowing layer. The balls with higher density at the bottom pushed the balls with
smaller density forward. It is worth to mention that from the simulation results, the velocities of
the blue balls at free surface is the largest, which result in that the balls with large size migrated to
the front of the flow. The segregation mechanism simulated in the numerical model is in consistent
with what is aforementioned in the physical model tests. Ashwood and Hungr (2016), Choi et al.
(2014), Choi et al. (2015), Kwan (2012), Lo (2000), Ng et al. (2014), Ng et al. (2017) have
investigated the impact forces on the barrier which is however not considered in the present study,
as this is not the main theme of the present work.

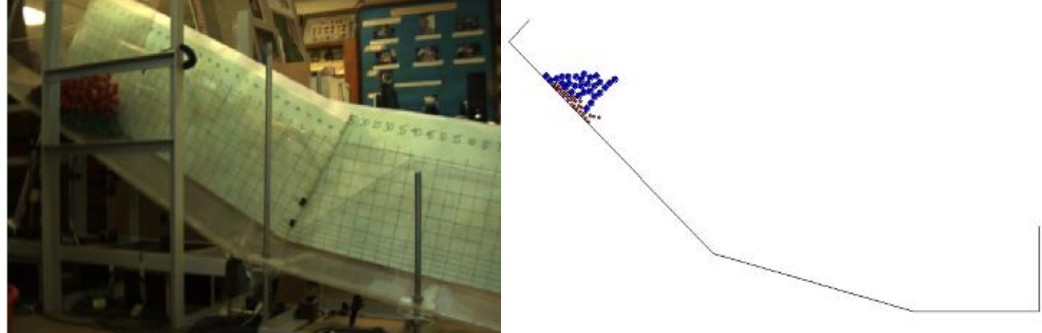


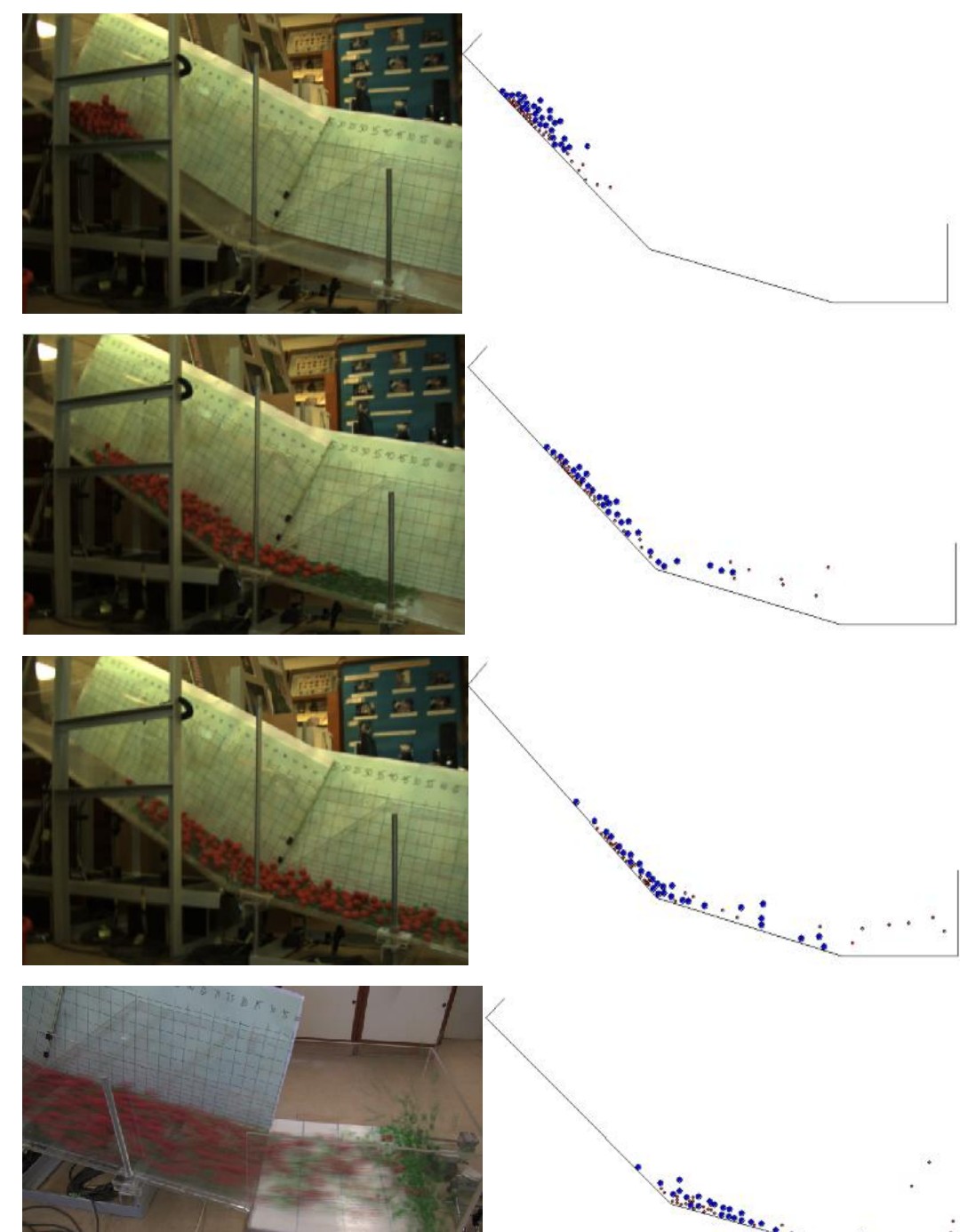





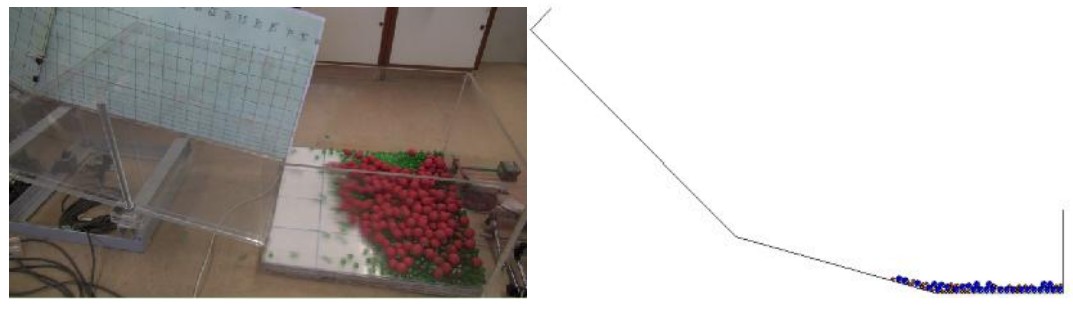

| Fig. 9a. Flow pattern of multi-size | Fig. 9. Flow pattern of multi-size |
| balls flow in physical test | balls flow in numerical test |

Fig. 9. Flow Pattern of multi-size particle flow composing of red plastic balls and green glass balls

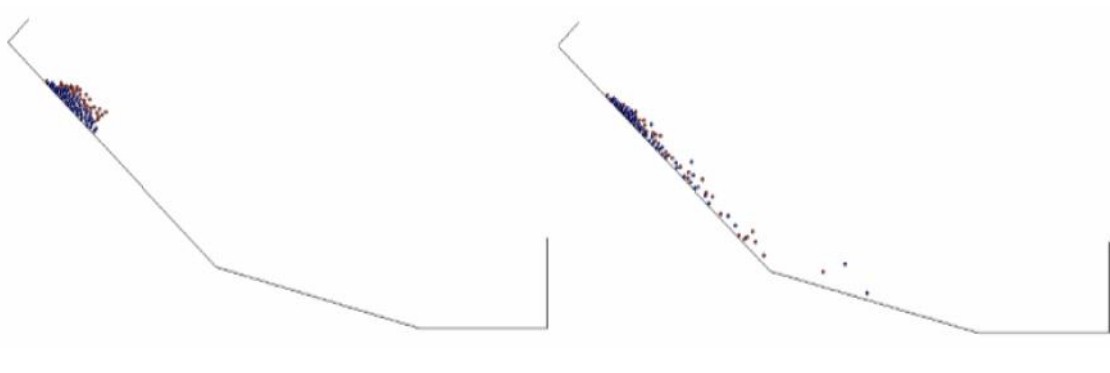

(a) Start of flow                              (b) 1/3 of flow time

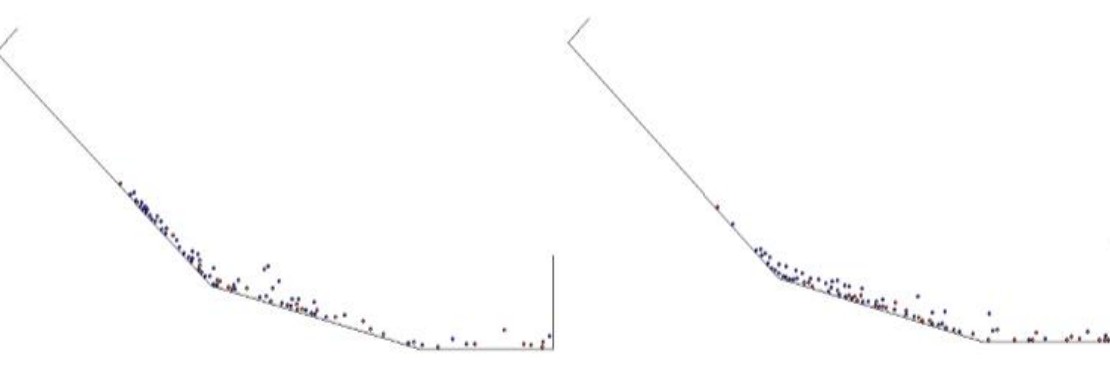

(b) 2/3 of flow time                           (d) end of flow

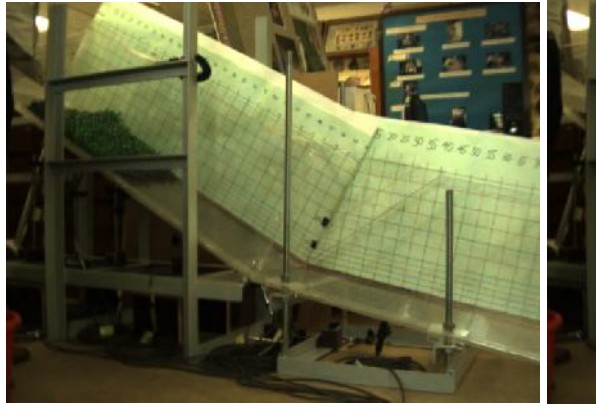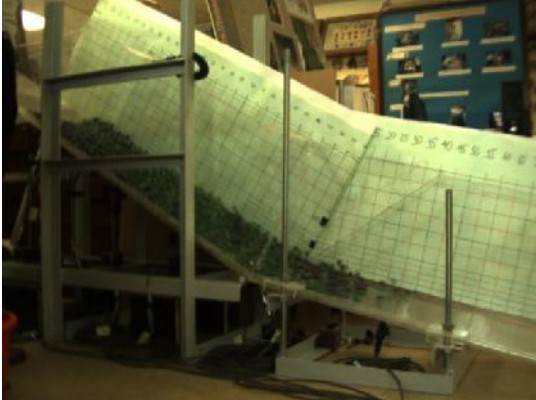


(e) Photo at start of flow             (f) photo at 1/3 of flow time

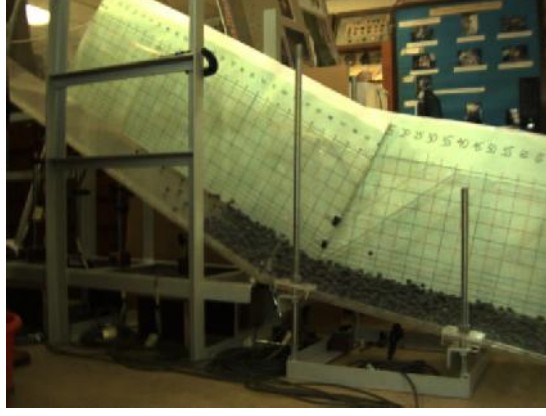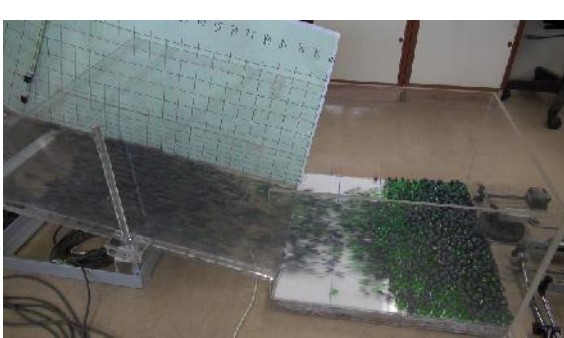


(g) photo at 2/3 of flow time            (h) photo at final stage


Figure 10. Flow Pattern of multi-size particle flow composing of black plastic

balls and green glass balls


**3.3 The effect of the flume jump**
To reduce the impact force and velocity of the granular flow mass, the authors have proposed to
add a jump in the flume as a pilot test in this study. From the results in this study, it is found that
the construction of a jump which has a very low cost has some small advantage in reducing the
impact from debris flow. Based on the present result, some rigid barriers in Hong Kong have
started to include a jump as a small benefit to the control of debris flow, and this is the reason for
carrying out such a test in the present research programme which is seldom considered in the past.
Figure 11 shows the numerical results of the flow pattern of the blue glass balls flowing on the
flume with or without a jump. The flow pattern of the blue glass balls flowing on the flume without
a jump in the numerical model is almost the same as the flow pattern of the red plastic balls in the
physical tests aforementioned. From the comparison of the flow pattern between Figure 11a and
Figure 11b, an important phenomenon was observed. The run up height of the balls flowing on the
flume with a jump is obviously lower than the run up height of the particles flowing on the flume
without a jump, which indicates that flume jump is able to facilitate the process of energy
attenuation and thereby has a good effect on suppressing the run up height of granular flow.
Figure 12 exhibits the velocity of the blue glass balls at different time step. In PFC2D, we have
developed the code to monitor the maximum velocity of the balls for comparison purpose, and the
monitored results are used to produce Fig.12. Black line represent the maximum velocity of the
blue glass balls with 10Kg weight flowing on the flume without a jump at different time step, while
the red line represent the same kind of balls with 13.55Kg weight on the flume with a jump. The
comparison of the velocities at point A and point B indicates that the peak velocity of the balls
flowing on the flume with a jump is pronouncedly smaller than that on the flume without a jump,
and the peak speeds of the balls on the flume with a jump were achieved earlier than balls on the
flume without a jump. It is worth to mention that the velocity of the balls is independent of the
mass of the test material, except that at the peak period.
Figure 13 shows the velocity profile of mono-size particles (blue glass balls) along the flume with
or without a flume jump. The length of the velocity vector represents the speed of the particles.
From Figure 13, it can be noticed that the front flow velocities are the largest compared with the
velocities of the particles at the rear of the flow. When these particles approached the lower part
of the flume, the velocity directions changed due to the difference of the flume angles. This is in
good agreement with the laboratory results mentioned above. Figure 13b shows that the velocity
of mono-size particles on the flume with a jump increased after the initial state. The largest flow
velocity was achieved at the moment when these particles intend to jump into the deposition zone.
The directions of flow velocities changed and the speed of particles decrease as soon as they fell
into the deposition zone. As with those particles moving on the flume with a jump, the velocity of
the particles flowing along the flume without a jump increased when they approached the
deposition zone, however, the velocity of these particles kept increasing when they flowed into the
deposition area and the peak speed was achieved just before the moment when they reached the
boundary of the deposition area. When the granular front impacted on the wall of the deposition
area, these particles at the front of the flow reflect back and collide with the following particles,
and that is the moment when the flow speed decelerated.
According to Figure 12 and 13, the peak velocity of the balls on the flume with a jump achieved
before they impacted on the wall of deposition zone compared with that without a jump, which is
meaningful to the engineers because the flume jump can effectively reduce the impact force on the
barrier.  Besides, the jump of the flume is capable of reducing the peak velocity of the dry granular
particle flow as well. To sum up, flume jump plays a useful role in attenuating granular flow,
therefore, flume jump is recommended to be applied in the design of debris flow barrier (which is
actually sometimes adopted in Hong Kong).






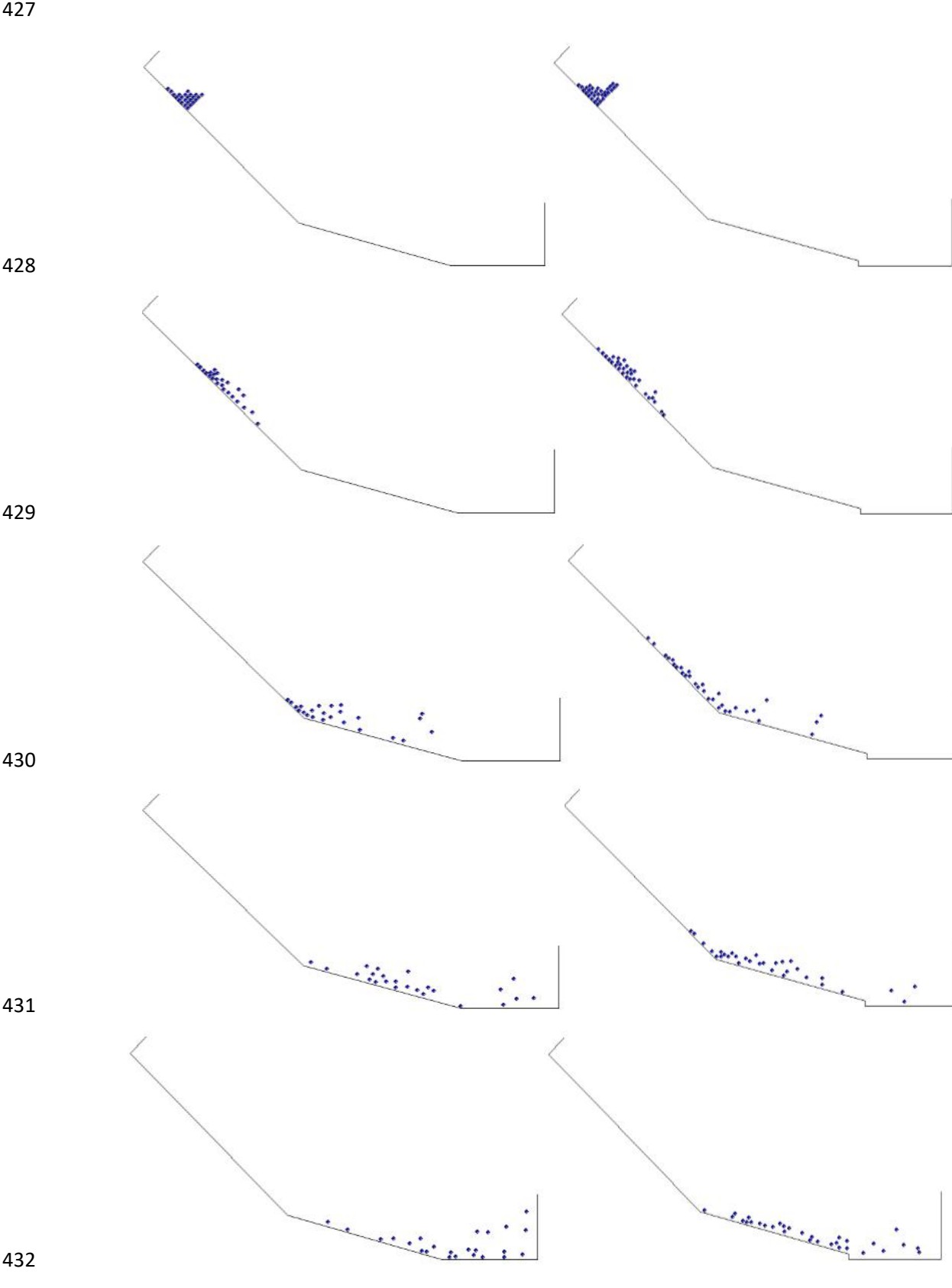




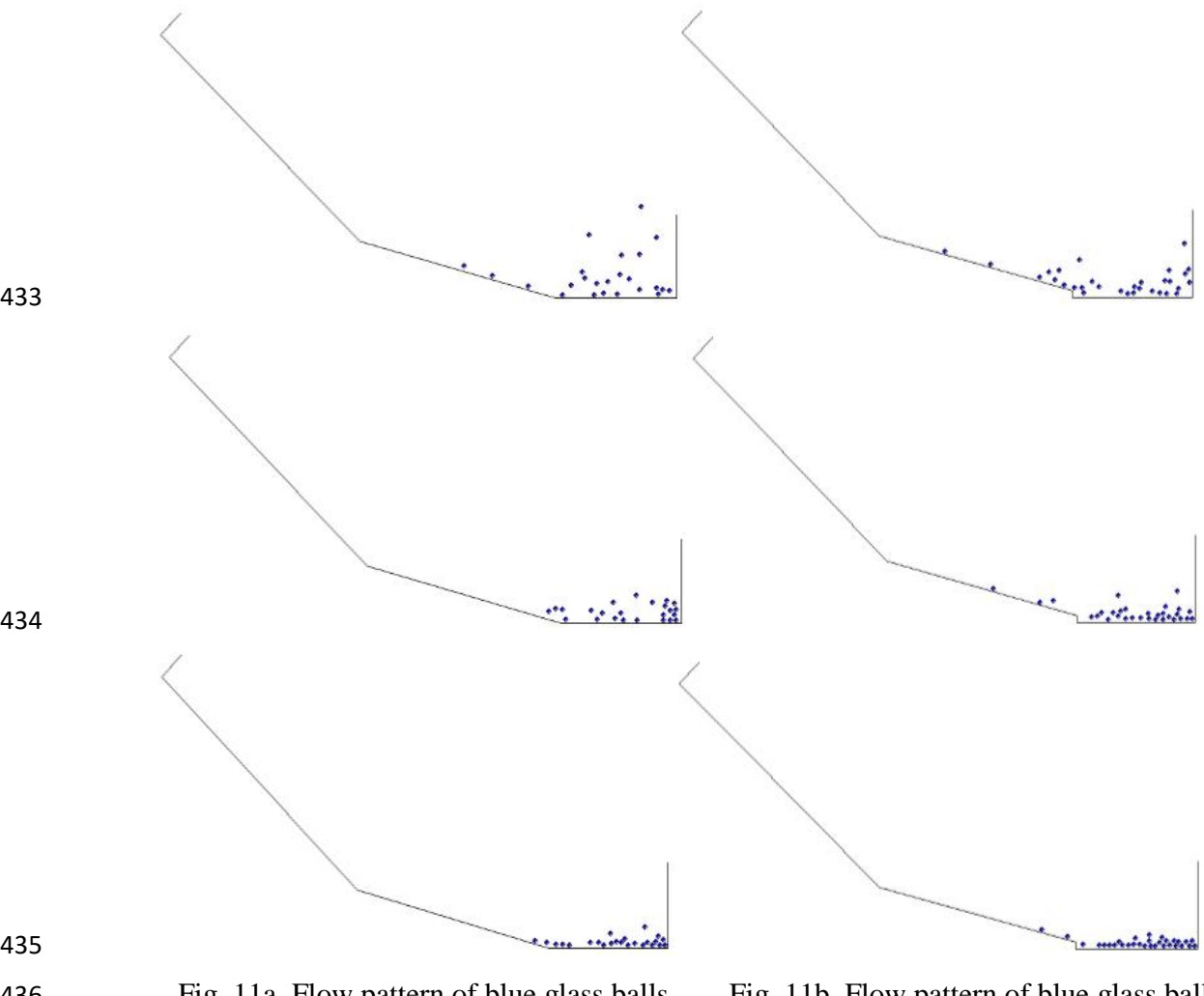

Fig. 11a. Flow pattern of blue glass balls          Fig. 11b. Flow pattern of blue glass balls

flowing along the flume without jump              flowing along the flume with jump

Fig. 11. Flow pattern of blue glass balls flowing on the flume with or without a jump


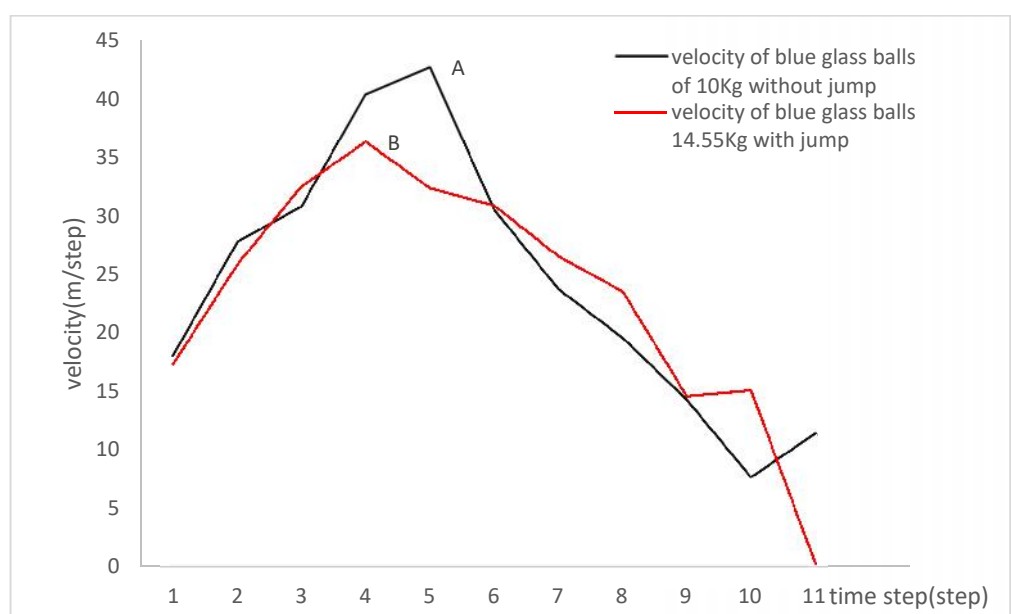


Fig. 12. Maximum velocity of blue glass balls in numerical model


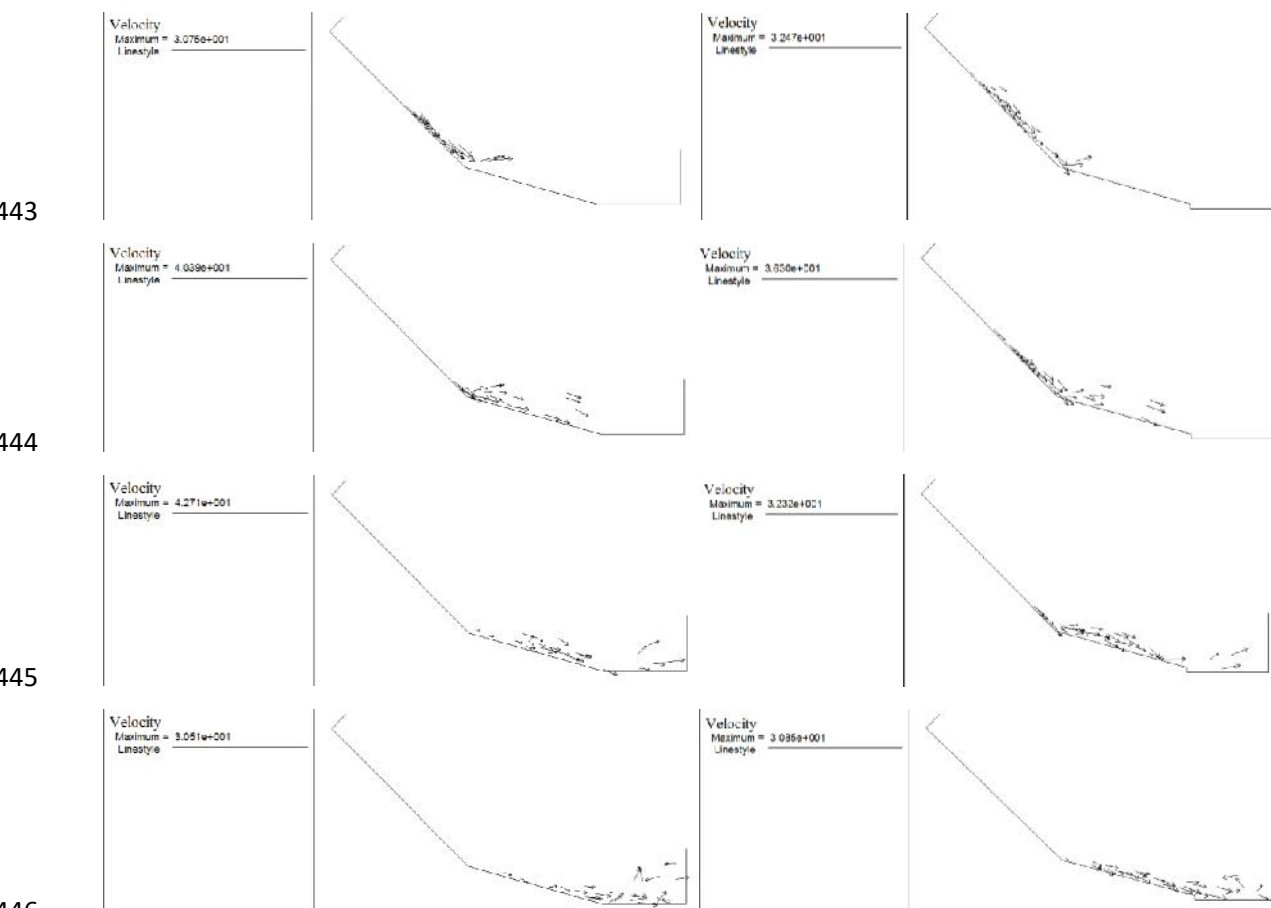





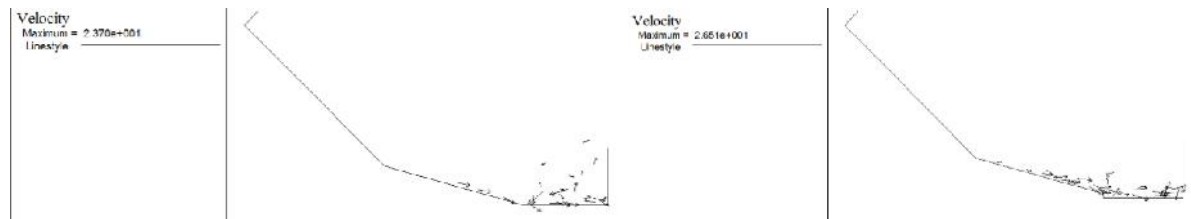

Fig. 13a. Velocity profile of balls on the flume    Fig. 13b. Velocity profile of balls on the

without a jump                                    flume with jump

Fig. 13. Velocity profile of blue glass balls in numerical model
It should be noted that the actual flow velocity of the balls can be traced back from the high speed
camera photos and the movie, but we do not present the results here because it is not the main
theme of the present study. Most importantly, DEM usually cannot give a good ~~for which~~
quantitative prediction unless the micro-parameters are fine tuned. The authors do not prefer such
tuning of the parameters, as such tuning cannot be performed before the tests. However, the
qualitative results from the DEM analysis and the laboratory tests are reasonable as found from
the present study, hence we can still accept the results from DEM in our discussion. Actually, the
authors have carried out limited tuning of the micro-parameters (not shown in this paper) in our
internal studies. Since the flow and segregation process are practically not affected by the change
of these micro-parameters (but the actual value of the flow velocity, run-out … are affected), we
have not included these results in the present paper, and the authors prefer to concentrate on the
segregation and jump for a flume test.

## 5. Large scale field tests

After the laboratory studies using a 1.5m long flume and glass/rubber balls, the authors have
carried out a large scale flume test which is shown in Fig.14. The flume is about 6m long, and 5
types of sand as shown in Fig. 15 are used in the field tests. The particle size within each type is
relatively uniform, and they ranged from 1-3mm, 3-5mm, 5-7mm, 7-8mm and above 8mm. The
friction angles for the 5 types of sand as determined from the deposition tests as shown in Fig.15b
are given by 28°, 30.3°, 29.1°, 31.5° and 33.7° respectively.

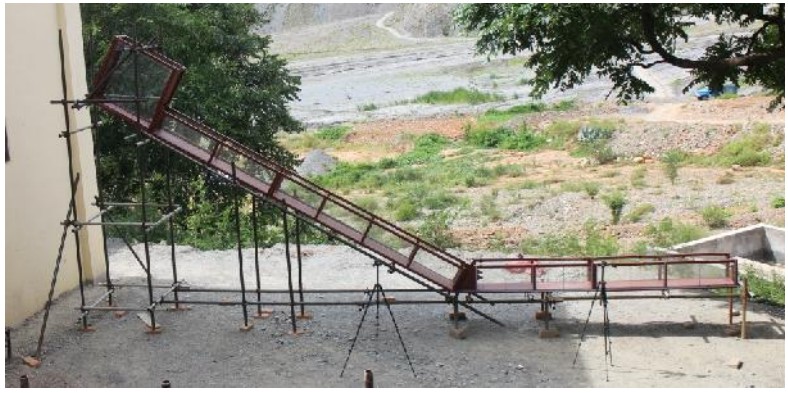


Fig.14 Large scale flume for field test

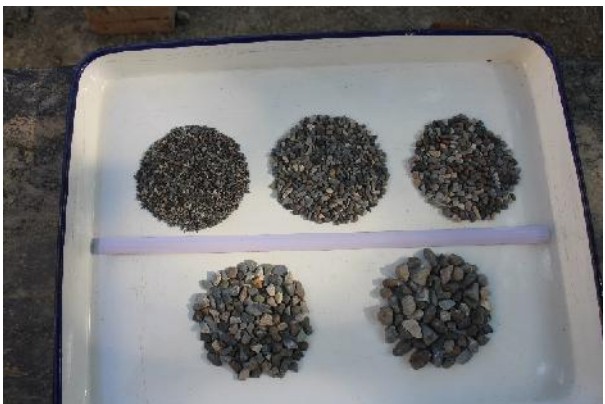 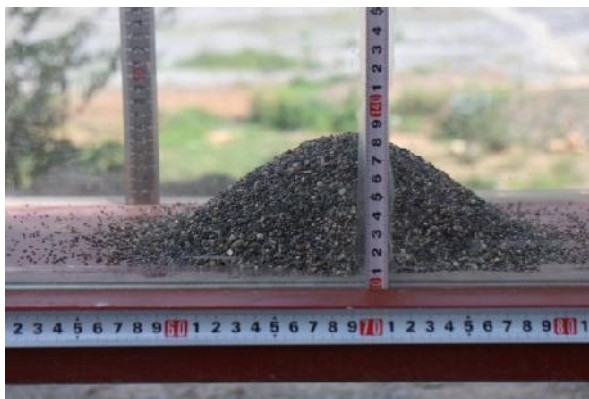


Fig.15a Sand used for granular flow tests  Fig.15b Deposition tests for sand

A series of tests with single, double and triple types of sand have been carried out, and only some
of the results are shown in this paper for comparisons with the laboratory tests. As shown in Fig.16,
the final deposition profile using type 1 (1-3mm) and type 4 (7-8mm) sands is shown. It is noticed
that the coarse grain sand move to the top of the flow, which are illustrated by Fig.17a to 17c. Such
results comply well with the laboratory studies. The control tests using coarse and finer sands are
shown in Fig.18. A closer look into the difference between Fig. 18a and Fig.16 is the profile at the
rear can reveal an important difference. For granular flow with 2 types of materials, the difference
in the height of deposit for the first meter as measured from the left is greater than that for the test
with single material (true for all single sand tests). Such phenomenon can be attributed to the effect
of the difference in the velocity flow between type 1 and 4 material, and type 1 material deposit at
the bottom during the flow. Based on the field tests, the importance of the particle size during the
segregation process as derived from the laboratory tests can be further verified.

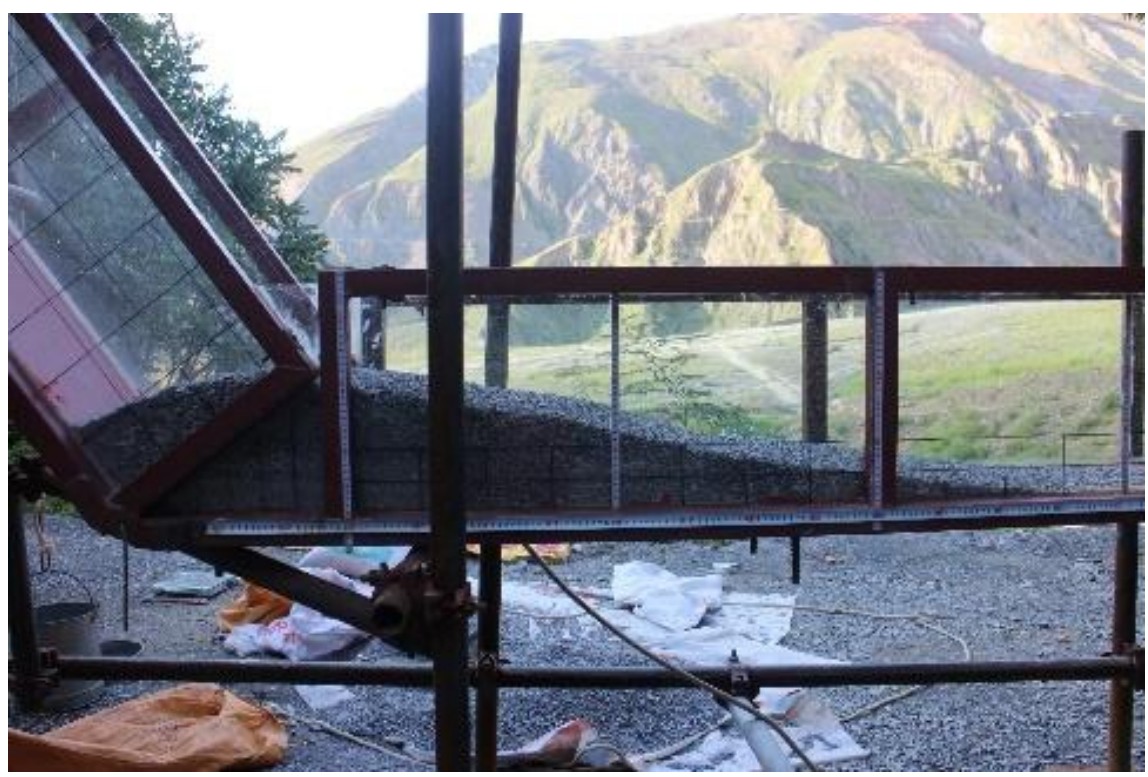
Fig.16 Final deposition after the granular flow for two materials (coarse and fine)

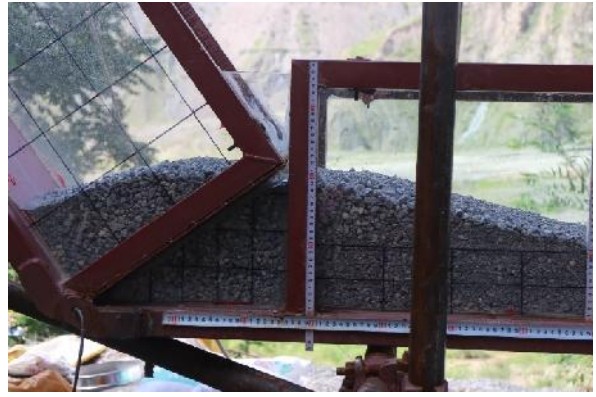  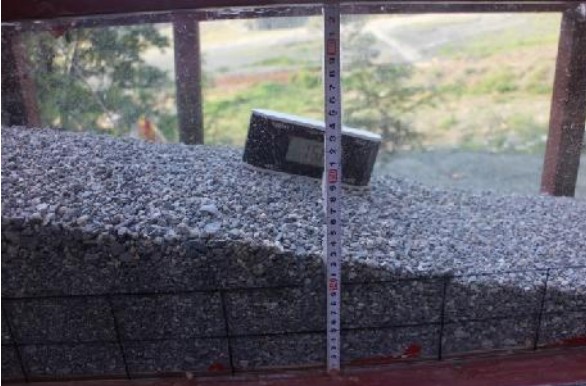


Fig.17 a Deposition at the rear of the deposit  Fig.17b Deposition at the front of the deposit

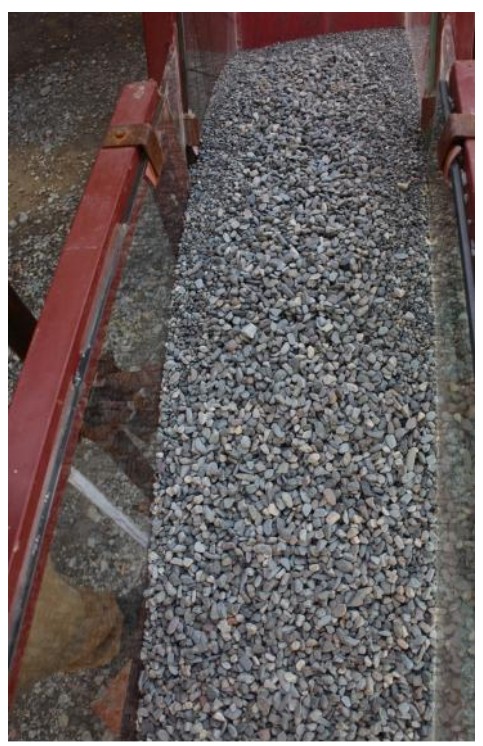


Fig.17c Front view of the deposition (2 materials)

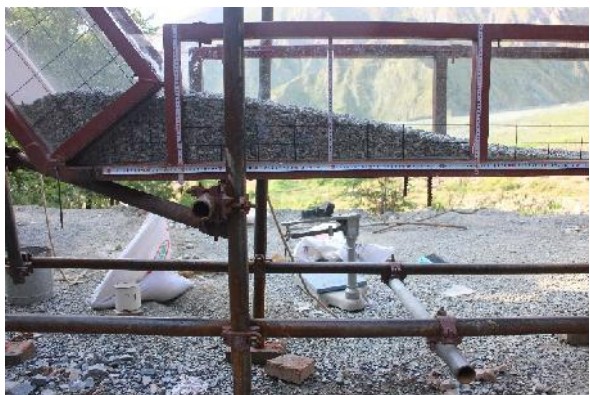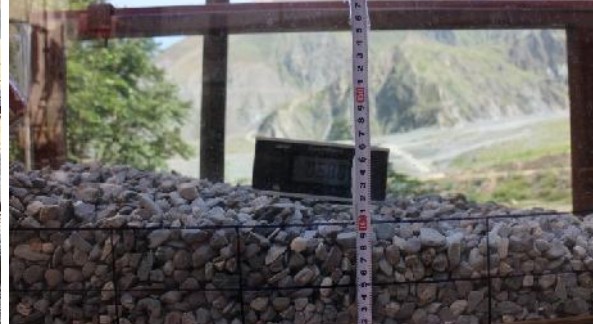


Fig.18a Front view of the deposition (type 4 material)  Fig.18b Close up view of the deposition

With reference to Fig.19, it is clear that the formation of the flow front, flow head, channelized
flow and levee from the present field test is very similar to that by Johnson et al. (2012). The
surface trajectories of the particles by Johnson et al. (2012) are also captured by the high speed
camera in the present laboratory and field tests. A coarse enriched surface layer has been obtained
by Johnson et al. (2012), and such phenomena are also obtained from the laboratory and field tests
and is clearly illustrated in Fig.17. Iverson (1997) has also found similar segregation from the
granular flow at Oregon (1996). It should be noted that for all the granular flow tests in the present
study, such segregation phenomenon is always obtained, as long as there are more than 1 materials
in the problems.
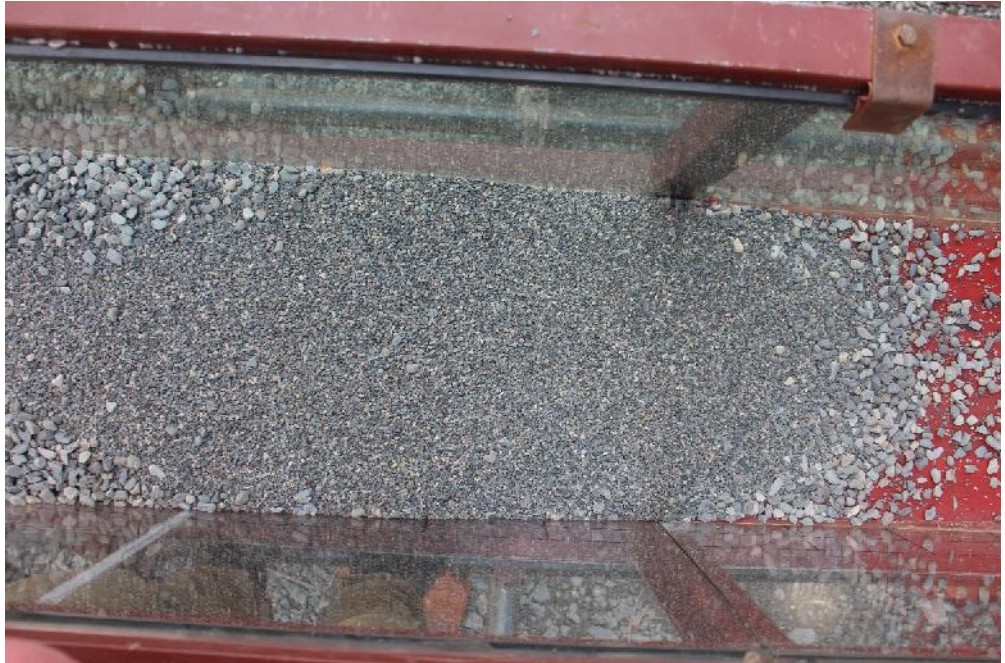
Fig.19 Front of the runout

## 6. Discussion

Laboratory tests were carried with numerical simulations through distinct element method to study
the flow pattern of dry granular flow. The study is important for the basic understanding of the
granular flow segregation problem and the importance of providing a jump in the flume or in the
actual protective measures. For the present tests, the flume base is even and smooth which result
in relative small dynamic frictional angle and less energy attenuation compared with the real
granular flow. Besides, the surfaces of the glass and plastic balls used in the experiments are
regular and smooth, while for debris flow occurring in nature, the debris materials are always
irregular and rough, which cause the dynamic internal frictional shear force between real scale
debris flow particles are relatively large with a lower and hence the run up height is lower. As a
consequence, the present tests is a conservative test to study the flow pattern of granular flow.
Such arrangement is necessary so as to separate the contribution of particle size distribution from
other parameters in the segregation process.

Physical tests were conducted to study the flow pattern of mono as well as multiple size particle
flows. In general, the results from the present study comply well with those from the literature.
Test results indicate that flow mass elongated under the action of shear force during the particles
flowed on the flume. For multi-size particles with different particle sizes, segregation always
occurs. Particles with larger diameters migrated upward and small particles moved downwards

because particles with smaller diameter can go through the gap between the larger particles. In addition, the density of the particle is another factor that play a role in the segregation process. Under the action of gravity, particles with higher density moved downwards faster and other particles with lower density were squeezed up. For the real scale debris flow, the debris material ranges from clay and silt to boulders while the differences in the densities between different types of particles are relatively small, hence particles size will be the most dominant factor which influence the segregation process. The top view from high speed camera indicates that the velocities of the large particles are higher than the velocities of the small particles. Granular particles with larger particles sizes travelled to the front of the flow where the velocities are higher. Larger particle size is observed to lead to a higher velocity. Such results are also in general agreement with the results by Takahashi (1980).

For the present work, the detailed movement of individual particle is hard to trace even with the help of high speed camera. Instead of that, the authors choose to trace the segregation process through the macro phenomena such as grain migration, segregation and the formation of the levee. Combined with the DEM analysis, the interpretation of individual grain movement as well as the formation of the segregation and levee can be assessed. Based on the various laboratory and field tests on flow with mixture of different material sizes, stiffness and density, it is established that the grain size distribution is the most critical factor in the flow process, as grain movement occur and control the flow process at about half of the flow process. The formation of the force chain which actually affect the flow process is also controlled by the grain size distribution. This result has an important implication in that most of the natural flow process involve debris of different grain sizes.

For the flow pattern of dry granular particles simulated through distinct element method, the simulation results of flow pattern are almost the same as the physical tests. Berger (2016), Chen and Lee (2000), Ghilardi et al. (2001) also obtained a reasonably well numerical modeling of the flow process for relatively simple flow problem which support the use of numerical analysis for the granular flow problem. In the present numerical model, a pronounced segregation process was observed as well, which comply well with many previous studies by Gray et al. (2003), Hákonardóttir et al. (2003), Iverson (1997), Johnson et al. (2012) and many others. Large particles went upwards while small particles went downwards. From the velocity vector figure, the velocities of the particles at upper layer as well as the velocities at the front of the flow were the largest. Savage numbers of the dry granular particles in present tests were larger than 0.1, which represent the collisional character of the flow. The flow behavior was hence more inertial than frictional. Flume jump have a significant influence on the impeding granular flow. When the particles flowed through the jump a large quantity of kinetic energy were consumed during this process. The peak velocities of particles flowing on the flume with a jump were lower than that without a flume jump. Besides, the peak velocities of the particles on the flume with a jump were achieved earlier, and after that the flow velocity started to decrease, which would make a great contribution for reducing the impact load. The run up height of the particles on the flume with a

jump was apparently lower than that without a jump. Thus, flume jump can help to reduce the flow velocity as well as suppress the run up height. In previous sections, detailed discussion about the formation of force chain from DEM are investigated, and such force chain has a major effect to the flow and segregation process which is actually observed from the tests. Without the DEM results, these phenomenon cannot be explained clearly. In this respect, the use of numerical modelling has provided an important help to the understanding of the flow and segregation process.

Comparing the physical and numerical test results, the macroscopic flow behavior in numerical models are consistent with the physical tests. Through a good selection of the model generation method and micro parameters, the distinct element method can produce a reasonable qualitative simulation of the behavior of dry granular flow for the consideration of the engineers. These results have useful contributions to the better understanding of the granular flow behavior which is not possible for the other classical methods. Up to the present, the engineers are still relying on some empirical methods such as dynamic impact earth pressure coefficient (Kwan 2012) or similar approaches for the design of flexible or rigid barrier, as granular flow process is complicated by many geotechnical and geographical complexities. The design of the barrier is still more an art than science up to the present, though some guidelines are available to help the engineers in the design. However, The DEM analysis in this study can supplement the field and laboratory studies for which the internal forces between the particles cannot be determined.

The flow process and segregation process from laboratory and field tests are similar in many respect – largely controlled by the particle size distribution. This is clearly illustrated from about 50 tests in our study. Limited photos are shown in this paper to limit the length of the paper. Thousands of photos and about a hundred movie files are obtained from the laboratory and field tests in this study, and only selected photos which are sufficient to illustrate the main purposes of the present work are shown in the present paper. The authors are however happy to share these materials upon request at ceymchen@polyu.edu.hk.

In the present paper, the effect of the flume inclination has not been investigated. Actually, the authors have carried out some tests on the effects of flume inclination. For the segregation process, the test results indicate that the basic conclusions from the present work remains unchanged, for practical purposes. Flume inclination has more important effects on the impact forces and erosion which are to be covered by the next stage of the present research work.

**7. Conclusion**

In the present study, two important phenomena in granular flow are studied. The first problem is the segregation process which is captured in all the tests in the present studies. The segregation phenomenon can affect the design of the barrier in different ways. The finer materials will be deposited at the bottom of the runout, and the relatively lower permeability of this layer will tend to drive the water level upward (somewhat similar to the perch water table phenomenon). This

may increase the destructive power of water. For the design of rigid barrier, the use of a suitable
water table will also be crucial to maintain adequate factor of safety of the barrier. Since
segregation will occur practically for majority of the debris flow problems, this effect should be
well studied and considered in the design of flexible and rigid barriers.

The authors have chosen flexible spherical rubber beads as well as rigid glass beads for the
laboratory, and the range of stiffness would be sufficient to cover most of the natural flow materials.
The segregation process as found from the laboratory test is actually similar to that in the field
tests using non-spherical sand. Through such selection, it is clearly demonstrated that particle size
distribution is a very critical factor in the segregation process, and it appears that it is more critical
than particle shape or stiffness.

To reduce the destructive power of the debris, a small jump in the flow channel is sometimes
applied in Hong Kong if the site condition allow. In general, the effect of this jump is small, and
is effective only for small volume debris flow which is the common case for Hong Kong.
Nevertheless, such provision can slightly reduce the destructive power of the debris. It is
interesting to note that there is virtually no study about the effect of the jump in the past, and the
present work provide some useful pilot works, for which more works may come out in the future.

One of the main limitations for the present study is that the flow material is limited to granular but
not cohesive material. The reason is that all debris flows in Hong Kong are practically granular
debris flows. The most critical factors in debris flow for Hong Kong include also different particle
size distribution (studied in the present work), topography and the effects of water. The present
work do not aim to consider all these effects simultaneously, but is confined to address the critical
issues as found in Hong Kong. Nevertheless, the present work will still be useful to many countries
where the flow material is mainly granular.

The authors are currently considering the next stage of field tests, for which the wet test will be
carried out (limited tests have been so far), and more equipment and measurements will also be
used.  Currently, the authors are constructing a laboratory flume where the base is rough. The
combined effect of base roughness and flume inclination angle will be carried out soon, and
hopefully the results will form the extension of the present paper. For the field test, most of the
researchers place a contained of wet sample and let the sample flow down. This approach is simple
to be executed, but the actual debris flow may not be like that. From the observations of several
debris flows in Hong Kong, the authors have noticed that erosion process is sometimes an
important phenomenon which is not simple to be reproduced in the field flume. The composition
of the flow material actually changes during the flow process. More thoughts will be given to the
setup of the wet field test in the future, and the base of the flume may be specially prepared with
some soil bedding to allow for erosion in the future tests.

**Acknowledgement**

The present project is funded from the Research Grants Council of the Hong Kong SAR Government through the project PolyU 152293/16E, and CityU University of Hong Kong Research Project No. 7004631, National Natural Science Foundation of China (Grant No. 51778313) and Cooperative Innovation Center of Engineering Construction and Safety in Shangdong Blue Economic Zone.

657

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
