# Peer review of "Laboratory and Field Test and Distinct Element Analysis of Dry Granular flows and segregation process"

_Natural Hazards and Earth System Sciences, 2018_

## Referee Comment (RC1) · Anonymous Referee #1 · 29 Mar 2018

The manuscript attempts to model the transportation process of debris flow using flume tests and DEM simulations on dry particles. Qualitative comparisons were compared between the flume tests and DEM models. The study confirms the contribution of particle size on dynamic segregation during transportation and deposition process and effectiveness of energy consumption of jump gaps before barriers. The technical contents are generally sound and the formality meets the technical writing requirements. The two weak points are no particle-fluid interactions and few quantitative comparisons. Specific comments are listed as followings.

1. The introduction section is too long. Because the trigger of debris flow is out of the scope of the manuscript, descriptions of triggering and cases should be shortened or removed.

[Figure]

2. Equations (1) & (2) should be removed because they are not related to the tests and simulations. Cited references should only cover important and relevant ones.

3. Figs. 4 and 5 can be removed or presented schematically. Fig. 6 is insignificant.

4. The procedure and details to determine frictional coefficients in Table 1 should be added for verifications. Only those tests related to current study should be included in this manuscript.

5. Descriptions of Lines 212-220 need quantitative evidence. For instance, the energy transformation, particle velocity patterns and deposition process need evidence to support the descriptions.

6. The determinations of parameters listed in Table 3 should be clearly described.

7. Velocity field or particle traces should be added in Fig. 10 to support the flow pattern descriptions.

8. The technique to construct Fig. 12 should be added to the content.

9. The large scale field tests should include both dry and wet tests to evaluate the significance of particle-fluid interactions and the confidence of implementing this study to real scenarios.

---

## Short Comment (SC1) · 17 May 2018

The authors present a set of flume experiments performed with dry mixtures of glass beads of various sizes, which were then modelled using a two-dimentional discrete approach. Moreover, the experiments explore the characteristics of the flow and deposition, but not of the movement initiation (as the material is simply released, rather than being destabilised through external forcing). So, I was wondering if the authors could modify the title, which at this stage is generic and too ambitious, to a more matching one such as "(qualitative comparison of) experiments and distinct element modelling of dry flows of granular mixtures" or similar.

Besides, the introduction from line 30 to line 73 seems unnecessary. Even though

it provides some contextualization and motivation for the authors' research work, it seems to actually constrain and limit it to a specific location. The issue of the correct modelling of the geo-hazards is universal, and the authors' study may be potentially applicable to other contexts. Hence, I suggest that the authors shift the focus in their introduction to the state of the art of their modelling approach (dry particle flow as opposite to continuum-based modelling) and provide the reader with an overview to understand the limitations of the existing modelling techniques, the motivation of the present work and how this work represents a step forward in our understanding and modelling capabilities. This is partly done from line 74 on. However, the motivation of the work does not stand out enough.

Lines 82-84 are just a list of citations. I suggest that the authors describe one of these classifications (if useful for the understanding of the manuscript) or leave this part out.

From line 89 to line 131: I think the authors could focus their literature review to experimental and theoretical experiences on dry flows first. Then, they could explain briefly how the presence of water changes the flow behaviour, if and how studies on dry flows are still applicable to wet flows, and why wet flows have not been studies in this work.

Lines from 132 to 143: the authors should motivate the choice of a 2D approach better. Even though flume test geometries attempt a 2D simplification, modelling them using a 2D granular approach may be not sufficient anyway, unless the modelling takes into account the changes of porosity (and not only) deriving from the use of discs that cannot move in the out of plane direction instead of balls that can do so in a 3D simulation. Moreover, current software and hardware allow for 3D computations of large assemblies of particles, so it is a pity that the authors did not explore this more realistic approach.

Lines 144 to 148: the authors could think of including the visual material (photos, videos) as supplementary material accompanying the paper, if they or the editor think it can be useful.

Lines 160-162: the authors could explain the advantages and limitations of using spherical glass beads instead of sand or gravel mixtures with generic shapes more in detail. For example, glass beads can rotate easily rather than simply sliding (which makes their apparent friction very small), while rotation is partly hindered when particles with irregular shapes are involved. Additionally, in real-size cases, particle crushing may occur during the movement, so grading will change during the flow and this may exert a feedback on its characteristics. Furthermore, I would be careful when interpreting the results of experiments in which the deposit is just one ball thick. Is this still representative of dry flows in nature or it is an oversimplification? Perhaps the authors could discuss on this.

Lines 221-222: it is anyway a pity that wet flows are not considered in the same work, as they would provide a much interesting insight and the article would become much more valuable. Moreover, it's generally the wet flows that are the most concerning ones in terms of hazard.

Lines 224-230: it would be good if the authors could explain their statement better and support it with data and/or references.

Line 285: model generation. A calibration of some parameters is involved here. The authors should specify which of the parameters have been calibrated by back analysis and show, for instance as a supplementary information, sensitivity analyses to the variation of these parameters.

Lines 333-335: I suggest that the authors provide quantitative metrics rather than a qualitative judgement of the agreement of the numerical simulations with the physical tests. These metrics could be geometric (e.g. runout distance, shape of the deposit), kinematic (e.g. flow velocity), dynamic (e.g. impact forces), energetic (e.g. energy dissipation, heat, acoustic emissions).

Line 448 onwards: the study of segregation in large flume tests seems very interesting. However, the results are shown and discussed only qualitatively (through photographs).

Again, I think the authors could discuss their observations in a more quantitative way (e.g. by studying the spatial changes of particle grading along the deposit and with depth, compared to the initial grading).

Lines 545 onward: this is of main concern. I think that providing only a "reasonable qualitative simulation of dry granular flow" without a quantitative insight is insufficient "for the consideration of the engineers". Also in the conclusions the authors provide only qualitative considerations. Therefore, I warmly encourage the authors to rethink their manuscript in a way that can provide quantitative results on which a solid scientific discussion can be based.

---

## Referee Comment (RC2) · Anonymous Referee #2 · 24 May 2018

The manuscript presents different approaches of characterizing particle flows down an incline: small-scale experiments with glass/plastic beads, large-scale experiments using granular material of sand/gravel, and the respective representation of the small-scale experiments using simulations based on Discrete Element Method (DEM) with the commercial code PFC2D. The motivation of the work is to provide an insight of the flow process and segregation of debris flows, although no consideration of fluid is present. Six types of particles have been used in the small-scale experiments, three of them are made of glass and the other three are made of plastic. 68 tests with a fixed inclination of 45 degrees have been carried with different mixtures of particles in the flowing mass: mono-disperse mixtures, mixtures with two types and also mixtures with three types, but no sensors where installed to measure flow velocities or depth. The

main investigation was concerning the segregation process and the role of particle size and density in it. Afterwards, 2-D numerical simulations where carried out to model the effect of segregation and also the presence of a jump with two types of particles, with no mention of the contact law or governing equations. The calibration process was not shown in the paper and only qualitative comparisons with the small-scale experiments were carried out. The presence of a jump was found to slightly modify the flowing velocity, leading to dissipation of kinetic energy of the flow. Finally, large scale experiments were carried out in longer flume with granular material consisting of sand/gravel with different sizes (five samples). A long qualitative description of the results claiming that the segregation process is well presence with same observations as the small-scale experiments (ex: inverse grading phenomenon). Discussions and conclusions are then presented. Some parts are written with a poor level of English which makes it hard for the reader to follow.

In order to decide on the publication of this paper in NHESS, I would like to highlight the following points:

First, the paper is titled 'debris flow', although a more appropriate name would be 'dry particle/granular flow', since no presence of fluid is considered. Such a presence would greatly influence the flow behavior and change its kinematics. The dry granular flow considered in the study, with the relatively big size of particles chosen could better resemble a rock avalanche than a debris flow. Moreover, the study focuses mainly on segregation process and effect of particle size, and this should be reflected in the title.

Second, the introduction mentions lots of statistics concerning the previous slope failures in China with no proper referencing (Lines 30-42). The same applies to the figures of previous events (e.g. Fig 1) where no reference is cited. In addition, when speaking about previous studies of debris flows, too much details are given that are unnecessary (e.g. the location of USGS flume). So many numbers are given concerning the geometry of previous flumes but no proper conclusions/open challenges of their work are presented (Lines 89-102). Previous work on modeling debris flow where detailed too

much with no added values (see for example equations 1 and 2 which are not used in the script afterwards), especially that these models where not based on DEM, which is the core of the present paper. On DEM studies, the authors failed to present a proper scientific literature review of the previous studies on granular flows modeling with DEM, and wrote instead a brief paragraph (Lines 132-137) on that with no highlight of what still needs to be done on this subject, Especially DEM simulations of segregation process.

For the small-scale lab experiments, the discerption of the carried out tests lack clarity and is found to be confusing for the reader (Lines 181-203). Furthermore, the quality of the images showing snap shots of the flow process is poor with no enough brightness, which makes it hard to drive strong conclusions. In addition, very often statements are made with no solid proof or measurements (e.g. lines 240-244 and lines 256-262). Such statements could be taken as assumptions to explain certain phenomenon but not as affirmative statements. More importantly and unfortunately, the experiments where carried out with no sensors to characterize flow depth and flow velocity (flow velocity could however be back calculated from the High speed camera).

More importantly, for DEM simulations, the section starts with mentioning studies on the run out which is not in scope of the paper (lines 286 – 288). In addition, no proper presentation has been given concerning the used contact law or the governing equations. Is it purely elastic? Elasto plastic? Elasto viscoplastic?. Particle are created in the model using the 'rain method' with no description of what it means: how are particles generated and at what time/condition do authors consider the sample to be in quasi-static condition and open the gate? Moreover, authors assume that their model is calibrated only by qualitative comparison with the experimental data. Very long description of the apparent 'agreement' between the model and experiment is detailed although such agreement is hard to judge because it is only qualitative and because of the low quality of experimental images. For a calibration to be justified, a more in-depth comparison with flow velocity and depth should have been carried out between model

and experiment. There is also no presentation of the most sensitive parameters of the model that needed calibration. Such a calibration process is crucial for the understanding of the model's results. It might be the case that same results could be obtained with more than one set of parameters. Strong arguments are presented concerning the flow regime and whether it is inertial or frictional, although no concrete measurements exist to calculate Froude number using velocity and height. In Fig 12, it is not clear which velocity is it (in flow direction, the norm of the velocity vectors .. etc). The only paragraph that is supported by quantitative measurements is in lines 400-415, although results of Fig 13 are hard to read due to its poor quality.

The large-scale experiments were also carried out with proper quantification of flow velocity and height through sensors. Authors depended on qualitative description of the segregation process which is harder to judge that the colored particles in the small-scale experiments. Authors claim that the results are similar to those of many previous studies, although it is not supported by quantifiable evidence.

The discussion part, which is supposed to take one step deeper in the analysis of results, included only a mixture of the abstract, the previously presented literature and a repetition of what have already been said in the previous section concerning the segregation phenomenon and the energy dissipating function of the jump. Claims concerning the savage number are not supported with measurements.

---

## Author Comment (AC1) · 6 Jul 2018

**Comments from reviewer**

The manuscript attempts to model the transportation process of debris flow using flume tests and DEM simulations on dry particles. Qualitative comparisons were compared between the flume tests and DEM models. The study confirms the contribution of particle size on dynamic segregation during transportation and deposition process and effectiveness of energy consumption of jump gaps before barriers. The technical contents are generally sound and the formality meets the technical writing requirements. The two weak points are no particle-fluid interactions and few quantitative comparisons.

Specific comments are listed as followings.

1. The introduction section is too long. Because the trigger of debris flow is out of the scope of the manuscript, descriptions of triggering and cases should be shortened or removed.

2. Equations (1) & (2) should be removed because they are not related to the tests and simulations. Cited references should only cover important and relevant ones.

3. Figs. 4 and 5 can be removed or presented schematically. Fig. 6 is insignificant.

4. The procedure and details to determine frictional coefficients in Table 1 should be added for verifications. Only those tests related to current study should be included in this manuscript.

5. Descriptions of Lines 212-220 need quantitative evidence. For instance, the energy transformation, particle velocity patterns and deposition process need evidence to support the descriptions.

6. The determinations of parameters listed in Table 3 should be clearly described.

7. Velocity field or particle traces should be added in Fig. 10 to support the flow pattern descriptions.

8. The technique to construct Fig. 12 should be added to the content.

9. The large scale field tests should include both dry and wet tests to evaluate the significance of particle-fluid interactions and the confidence of implementing this study to real scenarios.

**Reply**

The authors would like to thank the reviewer for the constructive comments, based on which a revised manuscript has been prepared to address the comments. Some of the questions as raised are not precisely the aim of the present study, and practically outside the capability of the computational technique/numerical model that is available up to the present. As mentioned in the conclusion of the revised manuscript: "The authors have chosen flexible spherical rubber beads as well as rigid glass beads for the

laboratory. The segregation process as found from the laboratory test is actually similar to that in the field tests using non-spherical sand. Through such selection, it is clearly demonstrated that particle size distribution is a very critical factor in the segregation process, and it appears that it is more critical than particle shape or stiffness." The main work from the present study is more on qualitative than quantitative, though we also aim to produce useful quantitative results from DEM, but this is not the main theme of the work, as this is usually achieved by tuning the micro-parameters (previous papers by the authors as well as many research papers on DEM).

The authors would like to reply the comments as follows:
1. The introduction part has been shortened to include more relevant materials only.
2. Eqs.(1) and (2) are removed. Thanks to the comments.
3. The authors personally view that Fig.4 and 5 can be retained. Some readers may want to see the actual photos of the flume and the end process. In particular, other reviewers do not have such comment, hence the authors prefer to retain these figures, unless there is a strong comment on it. Similarly, the authors view that Fig.6 is actually useful, as the photo give a clear view about the nature and texture of the balls.
4. The frictional coefficients in Table 1 are provided by the manufacturers, while the parameters in Table 3 are determined by in-house testing procedures. The testing procedures are provided in the section just before Table 3.

[Figure]
 rebound test

[Figure]
 Deposition test

 shear
force applied to determine the frictional coefficients between balls.

5. Line 210-220 – as mentioned in many places in the revised manuscript, due to the limitations of DEM, it cannot provide a good quantitative description of the flow process, up to the present. It is also not the purpose of the present manuscript to address this issue. However, the authors are also working on this issue to improve the quantitative assessment by using DEM, while the general procedures are to fine-tune the parameters until the computed results match well with the measured results. Personally, the authors do not favour this approach, as we can tune everything to match with experiments, even if the theory is wrong.

6. The determination of the parameters are based on laboratory tests, as given in the revised manuscript. A sample test report for the deposition test is given below, while the values as given in Table 3 are the mean values.

| Set | Balls | Angle (degree) | Fric |
|---|---|---|---|
| 1 | P(Black) | 27 | Average 0.365 |
| 2 | | 20 | |
| 3 | | 20 | Min 0.176 |
| 4 | | 20 | |
| 5 | | 10 | Max 0.577 |
| 6 | | 20-30 | |
| 7 | | 18.5 | |
| 8 | P(Red) | 20 | Average 0.429 |
| 9 | | 20 | |
| 10 | | 20 | |
| 11 | | 30 | Min 0.365 |
| 12 | | 23 | |
| 13 | | 20-25 | |
| 14 | | 30 | Max 0.577 |
| 15 | | 20-23 | |
| 16 | | 25 | |
| 17 | | 20 | |
| 18 | P(White) | 27 | Average 0.547 |
| 19 | | 30 | |
| 20 | | 28-30 | |

7.  The photos at different stages are added in Fig.10. We do not add in the velocity vector, as DEM cannot reproduce the results well quantitatively. The overall phenomenon is however reasonable.

8.  In PFC2D, we have developed the code to monitor the maximum velocity of the balls for comparison purpose. This is mentioned in the revised manuscript.

9.  We know that the large scale field should include the wet tests. There is however two very major limitations in the study : time and money. We need to pay for the workshop and the field equipment as well as various personnel working on site (the test site is in China, and further expenses are required for our research personnel's travelling and living allowances). Furthermore, we have only 1 month time in the test, from preparation to actual field tests. We are now trying to secure more fund for the next stage of works, for which the wet tests will be conducted.

---

## Author Comment (AC2) · 6 Jul 2018

**Comments from reviewer**

The authors present a set of flume experiments performed with dry mixtures of glass beads of various sizes, which were then modelled using a two-dimensional discrete approach. Moreover, the experiments explore the characteristics of the flow and deposition, but not of the movement initiation (as the material is simply released, rather than being destabilised through external forcing). So, I was wondering if the authors could modify the title, which at this stage is generic and too ambitious, to a more matching one such as "(qualitative comparison of) experiments and distinct element modelling of dry flows of granular mixtures" or similar.

Besides, the introduction from line 30 to line 73 seems unnecessary. Even though it provides some contextualization and motivation for the authors' research work, it seems to actually constrain and limit it to a specific location. The issue of the correct modelling of the geo-hazards is universal, and the authors' study may be potentially applicable to other contexts. Hence, I suggest that the authors shift the focus in their introduction to the state of the art of their modelling approach (dry particle flow as opposite to continuum-based modelling) and provide the reader with an overview to understand the limitations of the existing modelling techniques, the motivation of the present work and how this work represents a step forward in our understanding and modelling capabilities. This is partly done from line 74 on. However, the motivation of the work does not stand out enough.

Lines 82-84 are just a list of citations. I suggest that the authors describe one of these classifications (if useful for the understanding of the manuscript) or leave this part out.

From line 89 to line 131: I think the authors could focus their literature review to experimental and theoretical experiences on dry flows first. Then, they could explain briefly how the presence of water changes the flow behaviour, if and how studies on dry flows are still applicable to wet flows, and why wet flows have not been studies in this work.

Lines from 132 to 143: the authors should motivate the choice of a 2D approach better. Even though flume test geometries attempt a 2D simplification, modelling them using a 2D granular approach may be not sufficient anyway, unless the modelling takes into account the changes of porosity (and not only) deriving from the use of discs that cannot move in the out of plane direction instead of balls that can do so in a 3D simulation. Moreover, current software and hardware allow for 3D computations of large assemblies of particles, so it is a pity that the authors did not explore this more realistic approach.

Lines 144 to 148: the authors could think of including the visual material (photos,

videos) as supplementary material accompanying the paper, if they or the editor think it can be useful.

Lines 160-162: the authors could explain the advantages and limitations of using spherical glass beads instead of sand or gravel mixtures with generic shapes more in detail. For example, glass beads can rotate easily rather than simply sliding (which makes their apparent friction very small), while rotation is partly hindered when particles with irregular shapes are involved. Additionally, in real-size cases, particle crushing may occur during the movement, so grading will change during the flow and this may exert a feedback on its characteristics. Furthermore, I would be careful when interpreting the results of experiments in which the deposit is just one ball thick. Is this still representative of dry flows in nature or it is an oversimplification? Perhaps the authors could discuss on this.

Lines 221-222: it is anyway a pity that wet flows are not considered in the same work, as they would provide a much interesting insight and the article would become much more valuable. Moreover, it's generally the wet flows that are the most concerning ones in terms of hazard.

Lines 224-230: it would be good if the authors could explain their statement better and support it with data and/or references.

Line 285: model generation. A calibration of some parameters is involved here. The authors should specify which of the parameters have been calibrated by back analysis and show, for instance as a supplementary information, sensitivity analyses to the variation of these parameters.

Lines 333-335: I suggest that the authors provide quantitative metrics rather than a qualitative judgement of the agreement of the numerical simulations with the physical tests. These metrics could be geometric (e.g. runout distance, shape of the deposit), kinematic (e.g. flow velocity), dynamic (e.g. impact forces), energetic (e.g. energy dissipation, heat, acoustic emissions).

Line 448 onwards: the study of segregation in large flume tests seems very interesting. However, the results are shown and discussed only qualitatively (through photographs). Again, I think the authors could discuss their observations in a more quantitative way (e.g. by studying the spatial changes of particle grading along the deposit and with depth, compared to the initial grading).

Lines 545 onward: this is of main concern. I think that providing only a "reasonable qualitative simulation of dry granular flow" without a quantitative insight is insufficient "for the consideration of the engineers". Also in the conclusions the authors provide only qualitative considerations. Therefore, I warmly encourage the authors to rethink their manuscript in a way that can provide quantitative results on which a solid scientific

discussion can be based.

**Reply**

The authors would like to thank the reviewer for the constructive comments, based on which a revised manuscript has been prepared to address the comments. Some of the questions as raised are not precisely the aim of the present study, and practically outside the capability of the computational technique/numerical model that is available up to the present. As mentioned in the conclusion of the revised manuscript: "The authors have chosen flexible spherical rubber beads as well as rigid glass beads for the laboratory. The segregation process as found from the laboratory test is actually similar to that in the field tests using non-spherical sand. Through such selection, it is clearly demonstrated that particle size distribution is a very critical factor in the segregation process, and it appears that it is more critical than particle shape or stiffness." The main work from the present study is more on qualitative than quantitative, though we also aim to produce useful quantitative results from DEM, but this is not the main theme of the work, as this is usually achieved by tuning the micro-parameters (previous papers by the authors as well as many research papers on DEM).

The authors would like to reply the comments as follows:
1. The title of the paper has been revised to "Laboratory and Field Test and Distinct Element Analysis of Dry Granular flows and segregation process", which is better reflection of the actual works than the previous title. Thanks for the suggestion.
2. Lines 82-84 are just a list of citations. – true, but some readers may be interested to know more about this topic. I prefer to leave this small section in place, but I do have a strong view to keep this small section.
3. Line 144-148 - We have thousands of photos and about a hundred movie files for the laboratory and field tests. We need a portable hard disk to store all the materials, and only limited photos are shown in the paper for illustration. Most of the papers on debris flow are similar in this respect. The authors are happy to share these photos and movies upon request, but the selected photos are sufficient to illustrate the main purposes of the present work, and the authors have provided an email, through which interested readers can ask for the photos and movies for the present work.
4. Lines from 132 to 143 –It is true that three-dimensional distinct element modelling can be a better tool for the present problems, but the previous experience in three-dimensional distinct element modelling by the authors suggest that the amount of computer time can be significant. For the present study, the flume in both the laboratory and field tests are relatively narrow, and "off-track"

movement of the balls/grains are not major. In view of that, two-dimensional modelling has been adopted in the present study, and good results are actually obtained.

5. Lines 160-162 –We understand the problem of rotation, and that is the reason for the authors to publish several papers specially devoted to this problem in Distinct Element Analysis. We start the testing programme from a simple problem using spherical glass beads, which is easier for analysis of segregation. In our works for Distinct Element Analysis, we do create many special shapes for consideration. In a real problem, many factors can affect the run-out, and we hope to isolate out the effect of shape and devote our attention to grain size distribution only. After that, we perform the field tests using true sand and aggregate. The segregation phenomenon as we found for the smooth glass beads actually still apply for the true sand/aggregate in our tests !

6. Line 285: model generation - Except for the wall friction (should be small as the walls are relatively smooth) and wall stiffness, all the other parameters in Table 3 are determined by laboratory tests. For each parameter, five laboratory tests have been carried out, and the mean values are presented in Table 3. It should be noted that there is not a wide distribution in the laboratory determined parameters, hence the range of these parameters are not shown for clarity. Actually, we have carried out friction test, deposition test and rebound tests on the materials (sample photos included).

[Figure]
 deposition test

rebound test

7. Lines 333-335 – distinct element analysis is well known to be more suitable for qualitative than quantitative description. It is possible to tune the parameters so as to give quantitative matching, but this is not the purpose of the present work. Without test measurement, such matching is not possible. The purpose of the present work is to demonstrate the general applicability of the distinct element modelling in dry granular flow problem. For the tuning of the parameters to give quantitative comparisons, this is trivial and will not be discussed here, as this is not the main theme of the present work. (Please see the revised discussion). Due to the shape of the equipment, the run-out is stopped by an end plate so that run-out distance is not measured. The impact force is also not measured, as the main purpose of the laboratory test is the segregation during run-out and effect of the jump at the flume, and these have been mentioned in the abstract.

8. Line 448 onward – As mentioned before, DEM is more suitable for qualitative study so far. Quantitative study using DEM is still difficult, due to various difficulties in micro-parameters determination, contact model and other factors. These limitations are well known, and up to now are still open questions. The focus of the present paper is the segregation process from a qualitatively assessment, and the authors are also working on the possibility of quantitative DEM assessment so as to compare the computed results with the actual laboratory and field tests results on velocity of parameters and other information. It is mentioned in the discussion of the revised manuscript.

9. Line 545 onward – for the engineers, they need to assess the segregation process, run out distance as well as the impact forces on the barrier (which is not the main theme of the study). The authors have added a section in the discussion part "Up to the present, the engineers are still relying on some empirical methods such as dynamic impact earth pressure coefficient (Kwan 2012) or similar approaches for the design of flexible or rigid barrier, as debris flow process is complicated by many geotechnical and geographical complexities. The design of the barrier is still more an art than science up to the present, though some guidelines are available to help the engineers in the design.". Due to the nature of the problem, the design of barrier is still an art up to the present, and the present study try to provide some more understanding of the segregation process which are useful to the engineers. The present works do not try to address all the qualitative or quantitative assessments in debris flow problem.

**Comments from reviewer**

The manuscript presents different approaches of characterizing particle flows down an incline: small-scale experiments with glass/plastic beads, large-scale experiments using granular material of sand/gravel, and the respective representation of the smallscale experiments using simulations based on Discrete Element Method (DEM) with the commercial code PFC2D. The motivation of the work is to provide an insight of the flow process and segregation of debris flows, although no consideration of fluid is present. Six types of particles have been used in the small-scale experiments, three of them are made of glass and the other three are made of plastic. 68 tests with a fixed inclination of 45 degrees have been carried with different mixtures of particles in the flowing mass: mono-disperse mixtures, mixtures with two types and also mixtures with three types, but no sensors where installed to measure flow velocities or depth. The main investigation was concerning the segregation process and the role of particle size and density in it. Afterwards, 2-D numerical simulations where carried out to model the effect of segregation and also the presence of a jump with two types of particles, with no mention of the contact law or governing equations. The calibration process was not shown in the paper and only qualitative comparisons with the small-scale experiments were carried out. The presence of a jump was found to slightly modify the flowing velocity, leading to dissipation of kinetic energy of the flow. Finally, large scale experiments were carried out in longer flume with granular material consisting of sand/gravel with different sizes (five samples). A long qualitative description of the results claiming that the segregation process is well presence with same observations as the small-scale experiments (ex: inverse grading phenomenon). Discussions and conclusions are then presented. Some parts are written with a poor level of English which makes it hard for the reader to follow. In order to decide on the publication of this paper in NHESS, I would like to highlight the following points:

First, the paper is titled 'debris flow', although a more appropriate name would be 'dry particle/granular flow', since no presence of fluid is considered. Such a presence would greatly influence the flow behavior and change its kinematics. The dry granular flow considered in the study, with the relatively big size of particles chosen could better resemble a rock avalanche than a debris flow. Moreover, the study focuses mainly on segregation process and effect of particle size, and this should be reflected in the title.

Second, the introduction mentions lots of statistics concerning the previous slope failures in China with no proper referencing (Lines 30-42). The same applies to the figures of previous events (e.g. Fig 1) where no reference is cited. In addition, when speaking about previous studies of debris flows, too much details are given that are unnecessary (e.g. the location of USGS flume). So many numbers are given concerning the geometry of previous flumes but no proper conclusions/open challenges of their work are presented (Lines 89-102). Previous

work on modeling debris flow where detailed too much with no added values (see for example equations 1 and 2 which are not used in the script afterwards), especially that these models where not based on DEM, which is the core of the present paper. On DEM studies, the authors failed to present a proper scientific literature review of the previous studies on granular flows modeling with DEM, and wrote instead a brief paragraph (Lines 132-137) on that with no highlight of what still needs to be done on this subject, Especially DEM simulations of segregation process. For the small-scale lab experiments, the discerption of the carried out tests lack clarity and is found to be confusing for the reader (Lines 181-203). Furthermore, the quality of the images showing snap shots of the flow process is poor with no enough brightness, which makes it hard to drive strong conclusions. In addition, very often statements are made with no solid proof or measurements (e.g. lines 240-244 and lines 256-262). Such statements could be taken as assumptions to explain certain phenomenon but not as affirmative statements. More importantly and unfortunately, the experiments where carried out with no sensors to characterize flow depth and flow velocity (flow velocity could however be back calculated from the High speed camera). More importantly, for DEM simulations, the section starts with mentioning studies on the run out which is not in scope of the paper (lines 286 – 288). In addition, no proper presentation has been given concerning the used contact law or the governing equations. Is it purely elastic? Elasto plastic? Elasto viscoplastic?. Particle are created in the model using the 'rain method' with no description of what it means: how are particles generated and at what time/condition do authors consider the sample to be in quasi-static condition and open the gate? Moreover, authors assume that their model is calibrated only by qualitative comparison with the experimental data. Very long description of the apparent 'agreement' between the model and experiment is detailed although such agreement is hard to judge because it is only qualitative and because of the low quality of experimental images. For a calibration to be justified, a more in-depth comparison with flow velocity and depth should have been carried out between model and experiment. There is also no presentation of the most sensitive parameters of the model that needed calibration. Such a calibration process is crucial for the understanding of the model's results. It might be the case that same results could be obtained with more than one set of parameters. Strong arguments are presented concerning the flow regime and whether it is inertial or frictional, although no concrete measurements exist to calculate Froude number using velocity and height. In Fig 12, it is not clear which velocity is it (in flow direction, the norm of the velocity vectors .. etc). The only paragraph that is supported by quantitative measurements is in lines 400-415, although results of Fig 13 are hard to read due to its poor quality.

The large-scale experiments were also carried out with proper quantification of flow velocity and height through sensors. Authors depended on qualitative description of the segregation process which is harder to judge that the colored particles in the smallscale experiments. Authors claim that the results are similar to those of many previous studies, although it is not

supported by quantifiable evidence.

The discussion part, which is supposed to take one step deeper in the analysis of results, included only a mixture of the abstract, the previously presented literature and a repetition of what have already been said in the previous section concerning the segregation phenomenon and the energy dissipating function of the jump. Claims concerning the savage number are not supported with measurements.

**Reply**

The authors would like to thank the reviewer for the constructive comments, based on which a revised manuscript has been prepared to address the comments. Some of the questions as raised are not precisely the aim of the present study, and practically outside the capability of the computational technique/numerical model that is available up to the present. As mentioned in the conclusion of the revised manuscript: "The authors have chosen flexible spherical rubber beads as well as rigid glass beads for the laboratory. The segregation process as found from the laboratory test is actually similar to that in the field tests using non-spherical sand. Through such selection, it is clearly demonstrated that particle size distribution is a very critical factor in the segregation process, and it appears that it is more critical than particle shape or stiffness." The main work from the present study is more on qualitative than quantitative, though we also aim to produce useful quantitative results from DEM, but this is not the main theme of the work, as this is usually achieved by tuning the micro-parameters (previous papers by the authors as well as many research papers on DEM).

The authors would like to reply the comments as follows:

1. The title of the paper has been reworded to "Comparisons between Laboratory and Field Test and Distinct Element Analysis of Dry Granular flows" for clarity. There is however two very major limitations in the study : time and money. We need to pay for the workshop and the field equipment as well as various personnel working on site (the test site is in China, and further expenses are required for our research personnel's travelling and living allowances). Furthermore, we have only 1 month time in the test, from preparation to actual field tests. We are now trying to secure more fund for the next stage of works, for which the wet tests will be conducted. If fund is secured, we will go to the next step, and the new results will be presented under the title of debris flow.

2. The authors have removed many unnecessary case history/statistics, and have added the references for some of the given cases.

3. Eqs.(1) and (2) are removed. Thanks to the comments.

4. With reference to the DEM works, the authors would like to emphasize that the

most important result from the present study is that particle size distribution is the most important factor in the segregation process. The DEM analysis is carried out to illustrate that such phenomenon is also found from numerical analysis, but it is not the main theme of the work. The authors have published series of paper on DEM previously, and the limitations of DEM on quantitative study is well understood. In the revised manuscript, the authors have also included

5. We are sorry that some of the photos are not sharp enough, but qualitatively the flow and segregation process can be seen. To help the readers, we have provided the email address, through which the authors will provide the movie files for all interested readers.

6. The comments as given in lines 240-244 and 256-262 are based on that from DEM analysis, and this is mentioned in the revised manuscript. Just from the high speed camera and video, many internal phenomena are still difficult to be understood, and this is also the reason for carrying out DEM studies. Even though DEM cannot provide good quantitative results, qualitatively the results are still reasonable.

7. Flow velocity can be traced back from the high speed camera photos and the movie, but we do not present the results here because it is not the main theme of the present study. Moreover, due to the limitations of DEM to provide sufficient accurate quantitative results, we have no intention to present the flow vector in the first draft of the paper, but prefer to spend more effort on the segregation process. Every year, there are debris flows in Hong Kong, and segregation is always found. The title of the paper is also reworded to "Laboratory and Field Test and Distinct Element Analysis of Dry Granular flows and segregation process" to better reflect the main theme of this paper.

8. For the DEM analysis, the materials are taken as elastic, and the contact model is a simple elasto-plastic model based on that by Zohdi T.I. (2007) as given in PFC2D. It is mentioned in the revised manuscript. Sorry for not mentioning about these information.

9. The authors agree that an in-depth comparisons between flow velocity, depth, run-out and other results should be done to justify the agreement between DEM and test results. The authors have not done that because: (1) this is not the main theme of the present work, as mentioned previously; (2) a good matching can always be found by tuning the micro-parameters in DEM, which is the approach used by authors' papers and many other researchers' papers; (3) the authors have actually tried "limited" tuning in the micro-parameters, and qualitatively the same flow and segregation process are found. The authors are fully aware of the lack of quantitative study in the present work, and hopefully it will be the next step of

work, where water is added in the test. There is however two very major limitations in the study : time and money. We need to pay for the workshop and the field equipment as well as various personnel working on site (the test site is in China, and further expenses are required for our research personnel's travelling and living allowances). Furthermore, we have only 1 month time in the test, from preparation to actual field tests. We are now trying to secure more fund for the next stage of works, for which the wet tests will be conducted. We hope to carry out more detailed study about the flow process, and at that stage we will try to concentrate more on the quantitative study.

10. Referring to the parameters, many of the parameters are given in Table 3 are based on our friction tests, deposition tests and rebound tests, and sample photos are attached herewith

 deposition test

 rebound test

11. As mentioned previously, we have actually carried out limited tuning of the micro-parameters (not shown) in our internal studies. Since the flow and segregation are practically not affected by the change of the parameters (but the actual value of the flow velocity, run-out … are affected), we have not included these results in the paper, and discuss one of the most important issues which are relevant to that in Hong Kong-segregation.

12. The flow process and segregation process from laboratory and field tests are similar in many respect – largely controlled by the particle size distribution. We can see clearly about that, and only limited photos are shown to limit the length of the paper. Any interested reader can get the complete set of photos from the authors if necessary. Again, the authors have not attempted to compare the laboratory and field test results quantitatively, as they are not comparable directly.

13. We have not done anything on the Savage number, as this is not the main theme of the study.

---

## Author Response (AR2)

[revised manuscript text omitted]

**Suggestions for revision or reasons for rejection (will be published if the paper is accepted for final publication)**

The revision shows significant improvements in terms of technical content and clear presentation. Following comments shall be handled properly before it is accepted for publication.

1. In the response, the authors stated the wet tests were not performed due to the limitations of budget and time. Because the title has been changed, the content related to debris flow shall be changed accordingly. – The content has been greatly revised to reflect that the main theme is granular flow, which is a special case of debris flow. I have also used the term debris flow and granular flow carefully in the revised manuscript to reflect more precisely the works as discussed.

2. The caption of Fig. 2 is associated with slope stability analysis. However, the plot only shows the mesh. It shall be removed because it is not related to the "debris flow". I have added Fig.2b and some discussion to show the debris flow after the slope failure, to illustrate the relation between slope stability and debris flow, from engineer's view.

3. All the reviewers mentioned that in-depth analysis and discussions shall be presented. The authors shall address this issue and/or focus on the available findings. More discussion has been added to reflect the findings.

4. It is inadequate to copy the responses on the revision. The revised content shall meet the technical writing format and style. – Not all comments are copied and responded in the revised manuscript. I try to avoid this actually. Only those comments and responds which are useful to the reading and illustration of the paper are included in the revised manuscript. I have further revised the manuscript to improve the technical writing format. Some replies which are not that relevant to the paper are removed.

5. Section 3.3 is Irrelevant to the title. The authors shall explain or describe why this is included in the manuscript. – This is important in that a jump in the flume and the debris flow channel can help to reduce the impact force from the debris flow, which is seldom considered in the past. This has been found to be useful in Hong Kong, and this is now used in some of the rigid debris flow barrier in Hong Kong. This is further explained in the revised manuscript.

Furthermore, I have made various corrections and improvements to the use of English in the revised manuscript.

#